# Ligand bias underlies differential signaling of multiple FGFs via FGFR1

Kelly Karl[1], Nuala Del Piccolo[1], Taylor Light[1], Tanaya Roy[1], Pooja Dudeja[2,3], Vlad-Constantin Ursachi[2,4], Bohumil Fafilek[2,3,4], Pavel Krejci[2,3,4], Kalina Hristova[1]*

[1]Department of Materials Science and Engineering, Institute for NanoBioTechnology, and Program in Molecular Biophysics, Johns Hopkins University, Baltimore, United States; [2]Department of Biology, Faculty of Medicine, Masaryk University, Brno, Czech Republic; [3]Institute of Animal Physiology and Genetics of the CAS, Brno, Czech Republic; [4]International Clinical Research Center, St. Anne's University Hospital, Brno, Czech Republic

*For correspondence:
kh@jhu.edu

Competing interest: The authors declare that no competing interests exist.

**Abstract** The differential signaling of multiple FGF ligands through a single fibroblast growth factor (FGF) receptor (FGFR) plays an important role in embryonic development. Here, we use quantitative biophysical tools to uncover the mechanism behind differences in FGFR1c signaling in response to FGF4, FGF8, and FGF9, a process which is relevant for limb bud outgrowth. We find that FGF8 preferentially induces FRS2 phosphorylation and extracellular matrix loss, while FGF4 and FGF9 preferentially induce FGFR1c phosphorylation and cell growth arrest. Thus, we demonstrate that FGF8 is a biased FGFR1c ligand, as compared to FGF4 and FGF9. Förster resonance energy transfer experiments reveal a correlation between biased signaling and the conformation of the FGFR1c transmembrane domain dimer. Our findings expand the mechanistic understanding of FGF signaling during development and bring the poorly understood concept of receptor tyrosine kinase ligand bias into the spotlight.

## eLife assessment

This manuscript describes **useful** data on the mechanisms underlying the activation of the receptor tyrosine kinase FGFR1 and stimulation of intracellular signaling pathways in response to FGF4, FGF8, or FGF9 binding to the extracellular domain of FGFR1. **Solid** evidence for quantitative differences in the downstream responses induced by the three ligands is presented. This manuscript will be of interest to biochemists and cell biologists working on receptor tyrosine kinases and general cell signalling across membranes.

## Introduction

Fibroblast growth factor receptors (FGFRs) belong to the family of receptor tyrosine kinases (RTKs), which signal via lateral dimerization in the plasma membrane to control cell growth, differentiation, motility, and metabolism (*Lemmon and Schlessinger, 2010*; *Schlessinger, 2000*; *Paul and Hristova, 2019*). The four known FGFRs (FGFR1–4) are single-pass membrane receptors, with an N-terminal ligand-binding extracellular (EC) region composed of three Ig-like domains, a transmembrane (TM) domain, an intracellular (IC) region that contains a juxtamembrane (JM) domain, a kinase domain, and a cytoplasmic tail (*Goetz and Mohammadi, 2013*). The cross-phosphorylation of tyrosines in the activation loop of the FGFRs in the ligand-bound dimers is known to stimulate catalytic activity, resulting in the phosphorylation of secondary IC tyrosines, which recruit cytoplasmic effector proteins (*Eswarakumar et al., 2005*; *Karl and Hristova, 2021*). These effector proteins, once phosphorylated

by the FGFRs, trigger downstream signaling cascades that control cell behavior (*Schlessinger, 2004*; *Tomlinson et al., 2009*).

The FGFRs signal in response to EC ligands with a beta trefoil fold, called fibroblast growth factors (FGFs) (*Ornitz and Itoh, 2001*). As many as 18 different FGF ligands are known to bind to and trigger distinct cellular responses through a total of seven variants of four FGFRs (FGFR1–4) (*Ornitz et al., 1996*; *Zhang et al., 2006*; *Buchtova et al., 2015*). The diversity in FGF-FGFR interactions far exceeds that of other RTK systems, reflecting the morphogen role that FGF signaling plays in the development of many tissues and organs. FGFR1 is well known for its critically important role in the development of the skeletal system, and has been implicated in many cancers (*Passos-Bueno et al., 1998*; *Wilkie, 2005*; *White et al., 2005*; *Turner et al., 2010*; *Seo et al., 2014*). One specific example of a developmental process controlled by FGFR1 is limb bud outgrowth (*Mariani and Martin, 2003*; *Tabin and Wolpert, 2007*). The limb bud forms early in the developing embryo and consists of mesenchymal cells sheathed in ectoderm. The process starts as the mesenchymal cells begin to proliferate until they create a bulge under the ectodermal cells above. This is followed by the formation of the apical ectodermal ridge (AER) by the cells from the ectoderm, which specifies the proximal-distal axis of the limb, ensuring limb outgrowth (*Mariani and Martin, 2003*; *Tabin and Wolpert, 2007*). These roles are carried out by several FGF ligands, including FGF4, FGF8, and FGF9, which are produced and secreted by the ectodermal cells in the AER and have similar spatiotemporal expressions. The ligands diffuse into the mesenchymal region adjacent to the AER, and act upon the mesenchymal cells which express only the 'c' variant of FGFR1 (FGFR1c). This leads to the activation of signaling pathways downstream of FGFR1c, promoting survival and proliferation of the mesenchymal cells.

Experiments involving genetic ablation of FGF4, FGF8, and FGF9 have led to distinct limb malformations (*Mariani et al., 2008*). Thus, the actions of these ligands through FGFR1c are different, but the mechanism behind the differential response of FGFR1c to multiple ligands has not been investigated. More broadly, it is not known how a cell recognizes the identities of different FGF ligands, when bound to and signaling through the same FGFR. Here, we sought to investigate the molecular mechanism behind differences in FGFR1c signaling in response to FGF4, FGF8, and FGF9.

## Results

### Differences in FGF-induced activation of the ERK pathway in cultured chondrocytes

To investigate whether FGF4, FGF8, and FGF9 induce differential signaling via FGFR1c (referred to as FGFR1 here), we used rat chondrosarcoma (RCS) chondrocytes, an immortal chondrocyte cell line used to model proliferating chondrocytes in the developing limb (*Fafilek et al., 2018*). By western blot, the RCS cells express detectable amounts of both FGFR1 and FGFR2 (*Figure 1A*). While FGFR3 and FGFR4 cannot be detected by western blotting, transcripts for FGFR3 and FGFR4 can be found by qPCR. To create an RCS variant that expresses FGFR1 only, CRISPR/Cas9 was used to disrupt the endogenous FGFR2, FGFR3, and FGFR4 genes in RCS cells (*Kimura et al., 2021*) to generate cells expressing only the endogenous FGFR1 (RCS[Fgfr1]). As a control, we used CRISPR/Cas9 to disrupt all endogenous *FGFR* genes in RCS cells (*Figure 1A*; RCS Fgfr1–4 null). The RCS[Fgfr1] cells were treated with FGF4, FGF8, and FGF9 for up to 1 hr. The activation of the RAS-ERK MAP kinase pathway, which represents the major downstream signaling FGFR module, was monitored by western blot (*Figure 1B*, *Figure 1—source data 5*). FGF4, FGF8, and FGF9 showed significant differences in FGFR1-mediated activation of ERK, with FGF4 inducing the strongest signal, compared to weaker signals detected in cells treated with FGF8 and FGF9. To investigate the dynamics of ERK activation in further detail, we used a genetic reporter of ERK activity. Briefly, the RCS[Fgfr1] cells were transfected with the transcriptional reporter pKrox24 (*Gudernova et al., 2017*), engineered to induce the expression of eGFP upon ERK pathway activation. The pKrox signal was monitored for 48 hr by automated widefield microscopy. Significant differences in pKrox24 transactivation were found, with FGF4 inducing the strongest signal, in both magnitude and duration, compared to FGF9 and FGF8. Significant differences were also observed between FGF8 and FGF9-induced ERK activation (*Figure 1C*, *Figure 1—source data 6*). These experiments showed that the three FGF ligands induce differential activation of ERK in RCS chondrocytes, consistent with the genetic ablation experiments (*Mariani et al., 2008*), and prompted studies into the mechanism behind these differences.

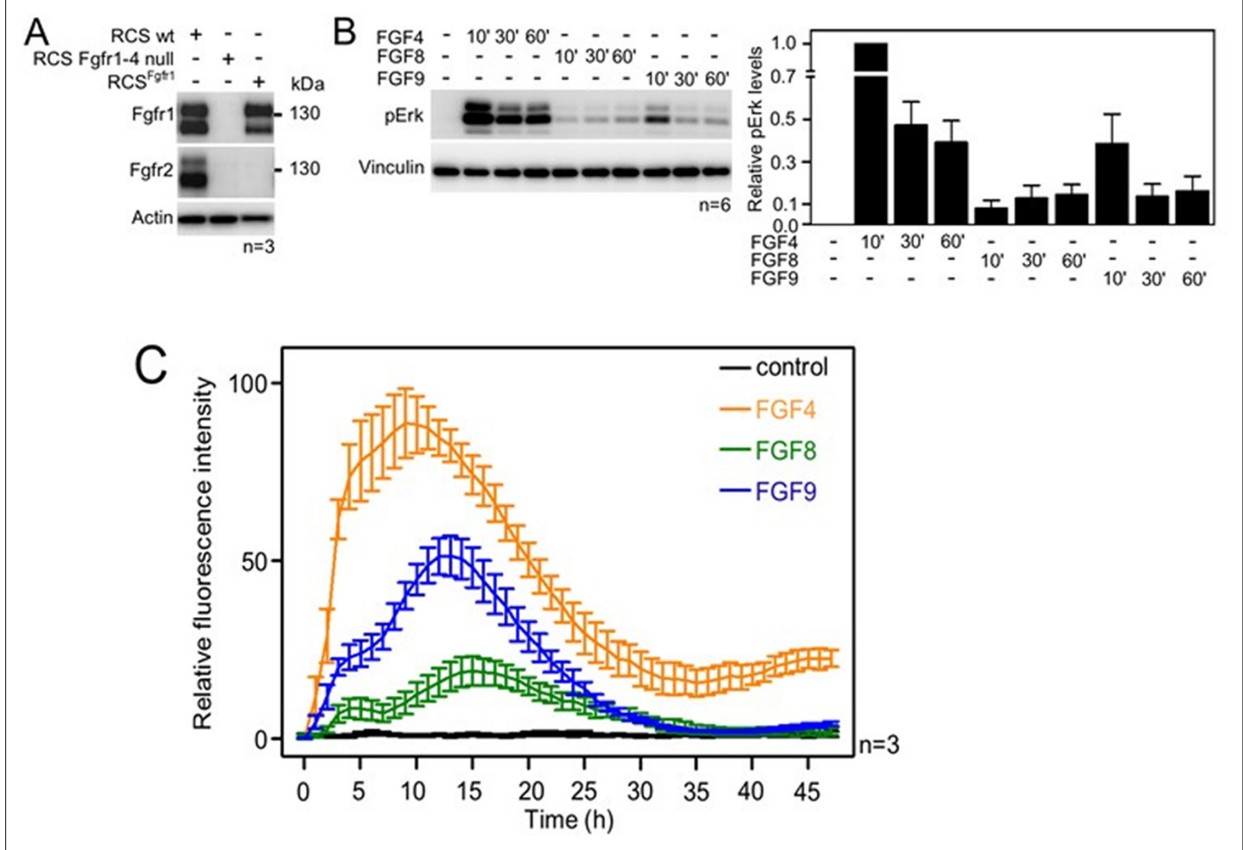

**Figure 1.** Activation of FGF signaling in RCS^Fgfr1 cells. (**A**) Expression of FGFR1 and FGFR2 in wild-type rat chondrosarcoma (RCS) cells, RCS cells *null* for FGFR1–4, and RCS cells expressing only endogenous FGFR1 (RCS^Fgfr1). Actin serves as a loading control; n, number of biologically independent experiments. (**B**) RCS^Fgfr1 cells were treated with FGF4, FGF8, and FGF9 for indicated times and ERK phosphorylation (pErk) was monitored by western blot. Vinculin serves as a loading control. pErk signal was quantified and graphed (right) as relative values compared to the 10' FGF4 stimulation; data show average and SEM of six biologically independent experiments. (**C**) RCS^Fgfr1 expressing the pKrox(MapERK)d1EGFP reporter were treated with FGF4, FGF8, and FGF9 and pKrox24 transactivation was monitored for 48 hr.

The online version of this article includes the following source data for figure 1:

**Source data 1.** Original files for the western blots in *Figure 1A*.

**Source data 2.** PDF containing *Figure 1A* and original western blots.

**Source data 3.** Original files for the western blots in *Figure 1B*.

**Source data 4.** PDF containing *Figure 1B* and original western blots.

**Source data 5.** Data plotted in *Figure 1B*.

**Source data 6.** Data plotted in *Figure 1C*.

## Differences in FGF-induced FGFR1 oligomerization

While most RTKs signal as dimers, it has been reported that under some conditions RTKs can form oligomers with different signaling capabilities (*Singh et al., 2018*). Therefore, we considered the possibility that FGF4, FGF8, and FGF9 promote the formation of different types of FGFR1 oligomers in the plasma membrane. We thus assessed the association state of FGFR1, labeled with YFP, using fluorescence intensity fluctuations (FIF) spectrometry. The fluorophores were attached to the C-terminus of FGFR1 via a GGS flexible linker; this attachment has been used before and has been shown to not affect function (*Sarabipour and Hristova, 2016*). The FIF experiments utilized 293T cells stably transfected with FGFR1-YFP.

FIF calculates molecular brightness of the YFP-tagged receptors in small regions of the cell membrane (*Stoneman et al., 2019*). The molecular brightness, defined as the ratio of the variance of the fluorescence intensity within a membrane region to the mean fluorescence intensity in this region, is proportional to the oligomer size (*Digman and Gratton, 2012*). The molecular brightness

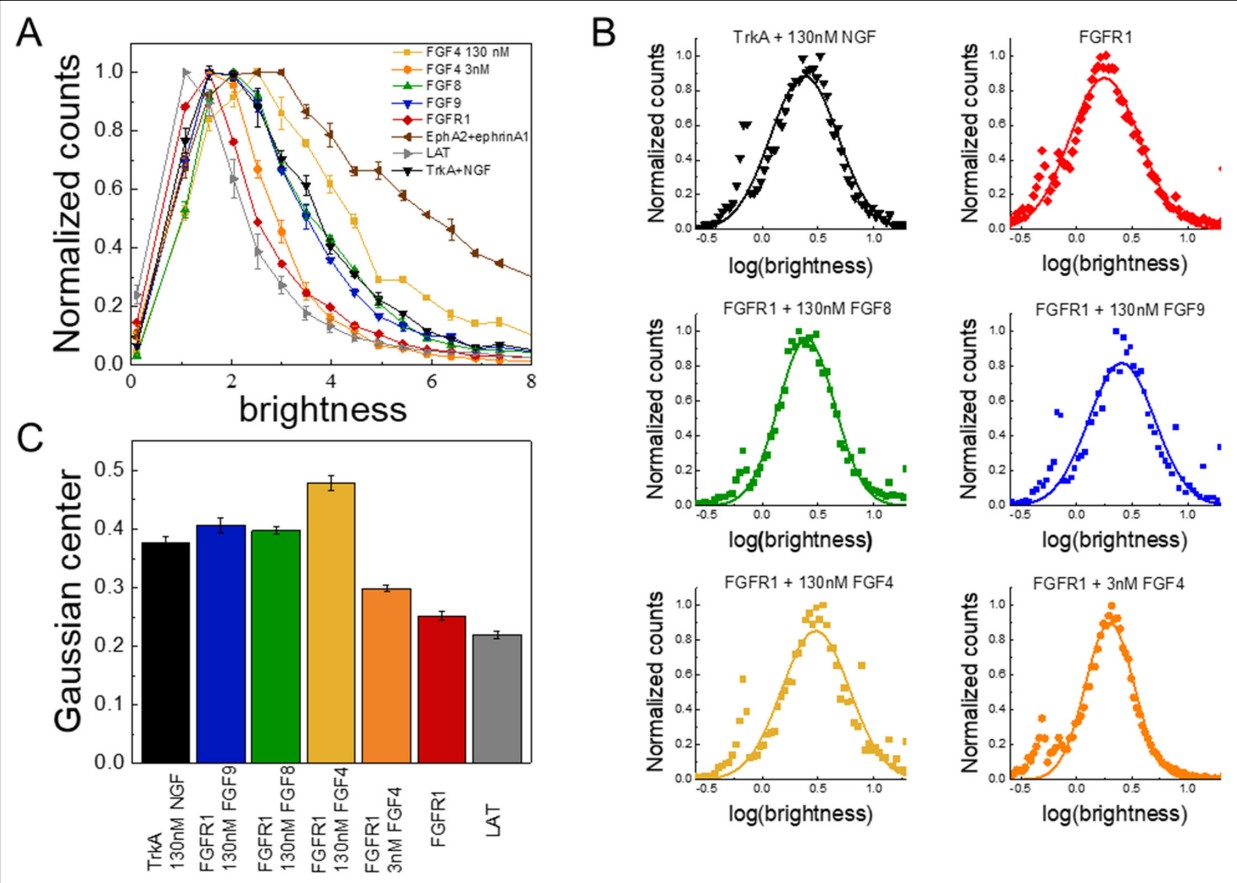

**Figure 2.** The oligomerization state of FGFR1, as measured by fluorescence intensity fluctuation (FIF) spectrometry. (**A**) Brightness distributions shown on the linear scale. Brightness scales with the oligomer size. Linker for activation of T-cells (LAT) (gray) is a monomer control, TrkA+130 nM nerve growth factor (NGF) (black) is a dimer control. EphA2 bound to ephrinA1-Fc (brown) is an oligomer control. All distributions are scaled to a maximum of 1. (**B**) Distributions of log(brightness). Points represent the experimental FIF data, and the solid lines are the best-fit Gaussians. (**C**) Means of the best-fit Gaussians and the standard errors of the mean. Each data set is derived from at least 100 cells in three biologically independent experiments.

The online version of this article includes the following source data for figure 2:

**Source data 1.** Fluorescence intensity fluctuations (FIF) brightness value distributions.

distributions of monomeric (linker for activation of T-cells [LAT]; gray) (*Paul et al., 2021*; *Paul et al., 2020*) and dimeric (TrkA in the presence of 130 nM nerve growth factor [NGF]; black) (*Ahmed et al., 2021*) controls in 293T cells are shown in *Figure 2A*; along with the brightness distribution for FGFR1 in 293T cells stably expressing FGFR1-YFP (red). We found that FGFR1, in the absence of ligand (red line), exists in a monomer/dimer equilibrium, as its brightness distribution is between the distributions of LAT (monomer control) and TrkA in the presence of saturating concentration of NGF (dimer control). The FIF experiments also report on the concentration of the receptors at the cell surface, which we find to be in the range 100–200 FGFR1/$\mu m^2$ in the stable cell line. This concentration is similar to previously reported FGFR expression levels of the order of ~80,000 receptors/cell, corresponding to ~80–100 receptors/$\mu m^2$ (*Moscatelli, 1987*).

Next, we performed FIF experiments in the presence of FGF4, FGF8, and FGF9. The FGFs were added at high concentrations (130 nM), which exceed the reported FGF binding constants (in the ~nM range) (*Roghani et al., 1994*; *Gleizes et al., 1995*) such that all FGFR1 molecules are FGF-bound. The brightness distributions in the presence of the FGFs are shown in *Figure 2A* (*Figure 2—source data 1*). We see that the brightness distributions recorded in the presence of FGF8 (green) and FGF9 (blue) overlap with the distribution for the dimer control, while the distribution for FGF4 (orange) appears shifted to higher brightness. In fact, these data fall between the dimer control and the large oligomer control (EphrinA1+ephrinA1-Fc) (*Singh et al., 2018*; *Seiradake et al., 2013*; *Seiradake et al., 2010*; *Nikolov et al., 2014*). This brightness distribution suggests that the average oligomer size may be

increased beyond a dimer in the presence of FGF4. We analyzed the likelihood of this possibility using a statistical test. Since the distributions of molecular brightness are log-normal, we analyzed the corresponding log(brightness) distributions which are Gaussian (*Figure 2B*). The parameters of the Gaussian distributions and the standard errors were calculated and used in a Z-test. Results, shown in *Figure 2*, show that there are statistically significant differences (Z>2, *Anderson et al., 2001*) between the FGFR1 brightness distribution in the FGF4 case and the distribution for the dimer control (TrkA+NGF). Likewise, there are statistically significant differences between the FGFR1+FGF4 brightness distribution, on one hand, and the FGFR1+FGF8 and FGFR1+FGF9 brightness distributions, on the other (Z>2). Further, the distributions measured for FGFR1+FGF8 and FGFR1+FGF9 are the same as the dimer control distribution. This analysis suggests that while FGF8-bound and FGF9-bound FGFR1 forms dimers, FGF4 binding may promote the formation of higher order FGFR1 oligomers.

## Differences in FGFR1 phosphorylation dose-response curves

Next, we studied FGFR1 signaling in response to FGF4, FGF8, and FGF9 using quantitative western blotting. In particular, we acquired FGFR1 dose-response curves while varying the concentrations of FGF4, FGF8, or FGF9 over a broad range. As we sought mechanistic interpretation of the results, we used the same 293T cell line used in the FIF experiments, since the FGFR1 expression and the FGFR1 oligomer size in the cell line are known.

The cells were incubated with FGFs for 20 min at 37°C, after which the cells were lysed, and the lysates were subjected to SDS-PAGE while probing with antibodies recognizing specific phospho-tyrosine motifs. We assessed several responses: (i) phosphorylation of Y653/654, which are the two tyrosines in the activation loop of the FGFR1 kinase that are required for kinase activity (*Goetz and Mohammadi, 2013*; *Schlessinger, 2004*; *Furdui et al., 2006*; *Lew et al., 2009*); (ii) phosphorylation of Y766 in the kinase tail of FGFR1, which serves as a binding site for PLCγ (*Sorokin et al., 1994*); (iii) phosphorylation of FRS2, which is an adaptor protein that is constitutively associated with FGFR1 through interactions that do not involve phosphorylated tyrosines (*Burgar et al., 2002*; *Gotoh, 2008*); and (iv) phosphorylation of PLCγ, which binds to Y766 (*Yang et al., 2016*). In addition, we blotted for the total expression of FGFR1, thus assaying for ligand-induced FGFR1 downregulation. Typical western blots are shown in *Figure 3A*. The band intensities from at least three independent experiments were quantified for each ligand concentration. The dose-response curves for different ligands were placed on a common scale by re-running some of the samples on a common gel, as shown in *Figure 3B*, and by following the scaling protocol in Materials and methods. The scaled dose-response curves are shown in *Figure 3C*; *Figure 3—source data 4–7*.

*Figure 3C* reveals unexpected differences in the shape of the dose-response curves. While the FGF8 and FGF9 dose-response curves appear sigmoidal when plotted on a semi-log scale, as expected for a rectangular hyperbolic curve (*Equation 1*), FGF4 exhibits an increase in phosphorylation up to 2.6 nM, followed by a marked decrease in phosphorylation for all studied responses with further increasing FGF4 concentration.

What could explain the very unusual FGF4 dose-response curves in *Figure 3C*? We first considered the possibility that the shape of the FGF4 dose response is due to strong FGF4-induced FGFR1 degradation at high FGF4 concentrations. Data in *Figure 3C*, however, do not support this view as the effect of FGF4 on FGFR1 downregulation is smaller when compared to the effects of FGF8 and FGF9. Thus, differential ligand-induced FGFR1 degradation cannot explain the shape of the dose-response curves.

We next considered the possibility that the shape of the FGF4 dose-response curve is due to differential FGFR1 de-phosphorylation kinetics. In particular, we asked whether at high FGF concentration, a fast de-phosphorylation occurs for FGF4-bound FGFR1, while the de-phosphorylation of FGF8-bound and FGF9-bound FGFR1 is slower. As the western blot data were acquired after 20 min of stimulation, such differences in kinetics could explain the shape of the dose response. We thus measured Y653/654 phosphorylation via western blotting as a function of time, when 130 nM FGF was added to the cells for a duration of 0, 1, 5, 10, 20, and 60 min. The averages of three independent experiments are shown in *Figure 4A*, along with the standard errors. We see a quick increase in phosphorylation over time, followed by a decrease which may be due to FGFR1 de-phosphorylation and/or FGFR1 downregulation. Notably, the phosphorylation decrease in response to FGF4 is very similar when compared to the decrease in the presence of FGF9, while the FGF8-induced phosphorylation decrease is smaller. Thus, the FGF4-induced de-phosphorylation kinetics are not fundamentally

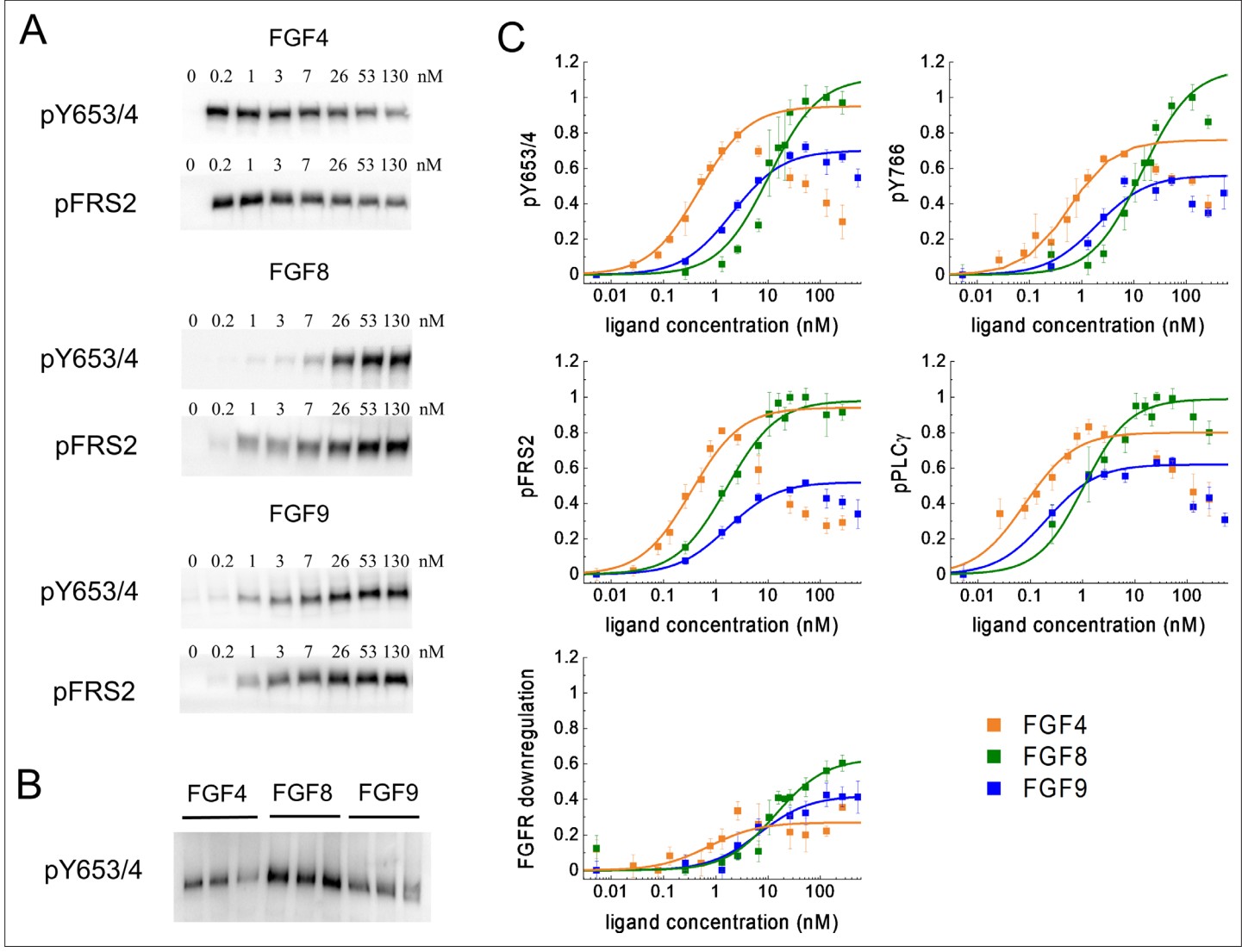

**Figure 3.** Phosphorylation of FGFR1 and downstream signaling substrates in HEK 293T cells. (**A**) Sample western blots for Y653/4 FGFR1 phosphorylation and FRS2 phosphorylation in response to FGF4, FGF8, and FGF9. (**B**) An example blot used for data scaling, where samples with maximum phosphorylation in response to FGF4, FGF8, and FGF9 are re-run on the same gel. (**C**) Dose-response curves from the western blot experiments. The points represent the averaged data, mean ± SEM, while the solid lines are the best-fit rectangular hyperbolic curves. Fit parameters are shown in *Table 1*. Data are from three to five biologically independent experiments.

The online version of this article includes the following source data for figure 3:

**Source data 1.** Original files for the western blots in *Figure 3A and B*.

**Source data 2.** PDF containing *Figure 3A and B* and original western blots.

**Source data 3.** pY653/654 phosphorylation dose-response curves.

**Source data 4.** Y766 phosphorylation dose-response curves.

**Source data 5.** pPLCγ phosphorylation dose-response curves.

**Source data 6.** pFRS2 phosphorylation dose-response curves.

**Source data 7.** Fibroblast growth factor receptor (FGFR) downregulation dose-response curves.

different when compared to the FGF9 results and cannot provide an explanation for the observed decrease in FGFR1 phosphorylation at high FGF4 concentration.

Another possible explanation for the shape difference in the dose-response curves could be that the FGF4-stabilized FGFR1 oligomers, observed in the FIF experiments at high FGF4 concentration (*Figure 2*), are less active than the FGFR1 dimers. To gain further insights into FGF4-induced FGFR1

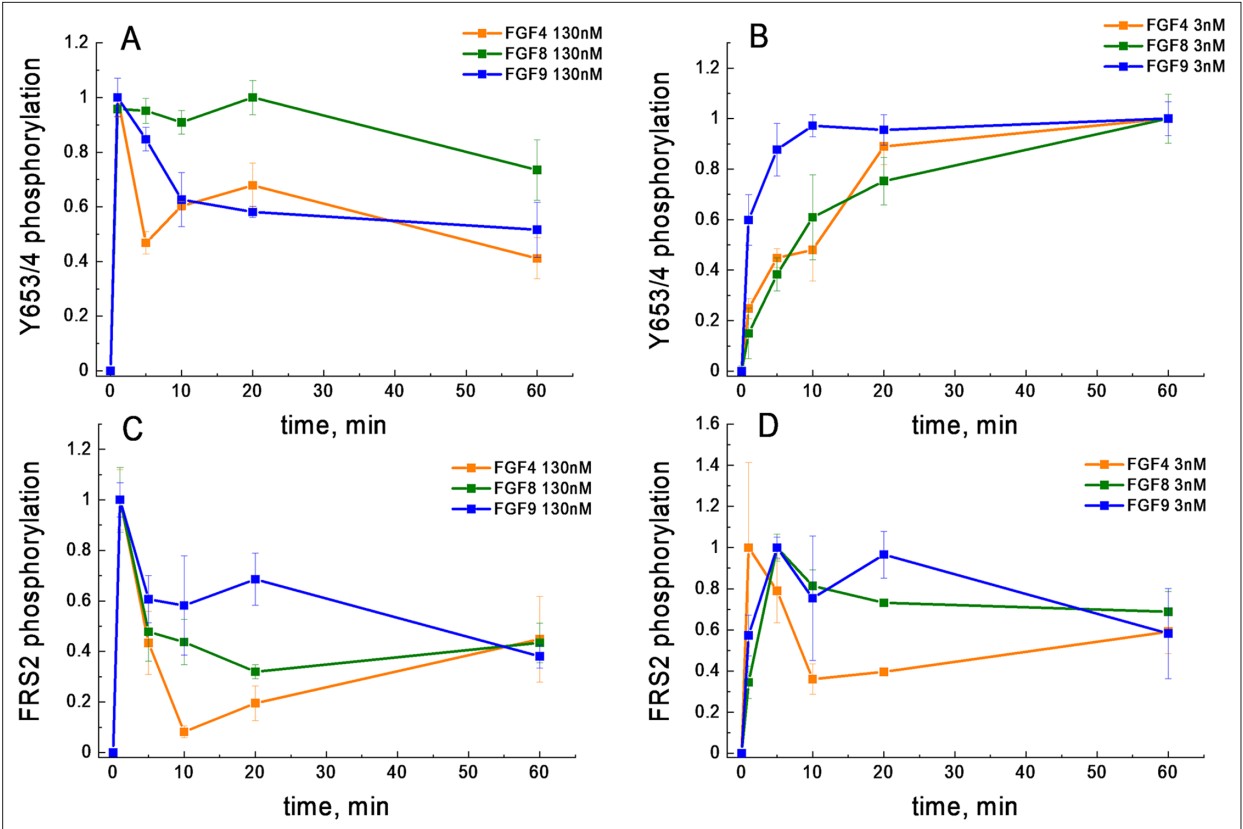

**Figure 4.** FGFR1 phosphorylation as a function of time after ligand addition. (**A**) Phosphorylation time course of Y653/654 at high ligand concentration (130 nM). (**B**) Phosphorylation time course of Y653/654 at low ligand concentration (2.6 nM). (**C**) Phosphorylation time course of FRS2 at high ligand concentration (130 nM). (**D**) Phosphorylation time course of FRS2 at low ligand concentration (2.6 nM). Shown are means and standard errors of replicates from three biologically independent experiments.

The online version of this article includes the following source data for figure 4:

**Source data 1.** All kinetics data plotted in *Figure 4*.

oligomerization, we performed FIF experiments in the presence of 2.6 nM FGF4, which corresponds to the peak of Y653/654 phosphorylation in the FGF4 dose-response curve. The brightness distribution at 2.6 nM FGF4, shown in *Figure 2*, lies between the monomer control and the dimer control. The Z-test analysis shows that the 2.6 nM FGF4 brightness distribution is significantly different from both the monomer and dimer control distributions, as well as from the distribution observed in the presence of high FGF4 concentration. Thus, at low FGF4 concentration, FGFR1 exists primarily in monomeric and dimeric states, while higher FGF4 concentrations (>2.6 nM FGF4) induce the formation of FGFR1 oligomers. The increase in phosphorylation at low FGF4 concentration can therefore be associated with FGFR1 dimers, while the subsequent phosphorylation decrease could be correlated with the appearance of oligomers in addition to dimers. Therefore, the assumption that FGFR1 oligomers are less active than FGFR1 dimers could provide an explanation for the observed shape of the FGF4 dose-response curves.

## Differences in phosphorylation potencies and efficacies, and demonstration of ligand bias

We next analyzed the dose-response curves in *Figure 3C* to determine the potencies and the efficacies of FGF4, FGF8, and FGF9. These two parameters were determined as optimal fit parameters in the context of rectangular hyperbolic dose-response curves (see *Equation 1*), as done in the G-protein coupled receptor (GPCR) literature (*Rajagopal et al., 2011*; *Kenakin and Christopoulos, 2013*; *Kenakin, 2016*). The efficacy ($E_{top}$ in *Equation 1*) is the highest possible response that can be achieved for a ligand, typically at high ligand concentration. The potency ($EC_{50}$ in *Equation 1*), on the other

**Table 1.** Best-fit parameters for dose-response curves in *Figures 3 and 5*.
$EC_{50}$ is the potency of the ligand, and $E_{top}$ is the efficacy (see *Equation 1*).

| pY653/4 | | | PLCγ | | |
|---|---|---|---|---|---|
| | $EC_{50}$, M | $E_{top}$ | | $EC_{50}$, M | $E_{top}$ |
| FGF4 | 4.77E-10±0.42E-10 | 0.95±0.03 | FGF4 | 7.98E-11±2.15E-11 | 0.80±0.04 |
| FGF8 | 1.04E-8±0.23E-8 | 1.11±0.07 | FGF8 | 1.05E-9±0.23E-9 | 0.99±0.03 |
| FGF9 | 2.09E-9±0.31E-9 | 0.70±0.02 | FGF9 | 2.05E-10±0.40E-10 | 0.62±0.02 |
| **pY766** | | | **pFRS2** | | |
| | $EC_{50}$, M | $E_{top}$ | | $EC_{50}$, M | $E_{top}$ |
| FGF4 | 5.84E-10±1.26E-10 | 0.76±0.05 | FGF4 | 3.42E-10±0.70E-10 | 0.94±0.06 |
| FGF8 | 1.42E-8±0.25E-8 | 1.16±0.07 | FGF8 | 1.62E-9±0.29E-9 | 0.98±0.03 |
| FGF9 | 1.94E-9±0.71E-9 | 0.56±0.05 | FGF9 | 1.65E-9±0.15E-9 | 0.52±0.01 |
| **FGFR1 Downregulation** | | | **Growth arrest** | | |
| | $EC_{50}$, M | $E_{top}$ | | $EC_{50}$, M | $E_{top}$ |
| FGF4 | 7.34E-10±4.54E-10 | 0.27±0.04 | FGF4 | 2.59E-12±1.1E-13 | 0.99±0.00 |
| FGF8 | 1.39E-8±0.39E-8 | 0.63±0.05 | FGF8 | 1.60E-9±2.3E-10 | 1.14±0.04 |
| FGF9 | 6.80E-9±2.08E-9 | 0.42±0.02 | FGF9 | 5.89E-11±8.5E-12 | 0.95±0.00 |
| **Collagen type 2 loss** | | | | | |
| | $EC_{50}$, M | $E_{top}$ | | | |
| FGF4 | 1.24E-11±2.79E-12 | 1.02±0.01 | | | |
| FGF8 | 8.51E-11±2.86E-11 | 0.77±0.03 | | | |
| FGF9 | 5.67E-11±2.89E-11 | 0.86±0.06 | | | |

hand, is the ligand concentration that produces 50% of the maximal possible response for a given ligand. A highly potent ligand will evoke a certain response at low concentrations, while a ligand of lower potency will evoke the same response at much higher concentration.

The fitting of the FGF4 dose-response curves presented a particular challenge, as we could only fit the increasing portions of the dose-response curves to a rectangular hyperbolic curve. As described in Materials and methods, we truncated the data for the fit, including the data in the ascending portions of the curves and taking into account that the average errors in the western blots are 5–10%. In particular, we truncated all the acquired dose-response curves at the highest ligand concentration that is within 10% of the maximum response value. Under the assumption that the decrease in the FGF4 dose-response curve is due to oligomerization at high ligand concentration, the best-fit FGF4 dose-response curves represent the response of the FGFR1 dimers to FGF4.

The best fits for all dose-response curves are shown in *Figure 3C* with the solid lines. The best-fit values of $E_{top}$ and $EC_{50}$ for all studied responses are shown in *Table 1*. FGF4 exhibits the highest potency, followed by either FGF8 or FGF9, dependent of the particular response. FGF8 exhibits the highest efficacy, followed by either FGF4 or FGF9, dependent of the particular response. In most cases, FGF8 is a full agonist, while FGF9 and FGF4 are partial agonists. However, FGF4 and FGF8 appear to be both full agonists when FRS2 phosphorylation is probed (see *Table 1* and *Figure 3C*).

The results in *Table 1* show that the rank-ordering of the different ligands is different when different responses are probed, which is indicative of ligand bias. 'Biased agonism' or 'ligand bias' is the ability of different ligands to differentially activate different signaling pathways downstream of the same receptor (*Gundry et al., 2017*). Ligand bias reflects not just quantitative differences in downstream signaling, but a fundamental difference between the signaling responses to different ligands (*Kenakin, 2019*; *Ehlert, 2018*). To determine if bias exists or not, we calculate bias coefficients using *Equation 2*. Each bias coefficient, shown in *Table 2*, compares two responses and two ligands and reveals

**Table 2.** Calculated bias coefficients using *Equation 2*.
Gray shading indicates statistical significance between either FGF4 or FGF9 and the reference ligand FGF8 (see *Supplementary file 4* for p-values).

|  | β | |
|---|---|---|
|  | 4v8 | 9v8 |
| pY653/4 vs pY766 | –0.07±0.16 | 0.05±0.15 |
| pY653/4 vs PLCγ | –0.24±0.18 | 0.01±0.14 |
| pY653/4 vs pFRS2 | –0.61±0.16 | –0.78±0.12 |
| pY766 vs pPLCγ | –0.18±0.19 | –0.04±0.17 |
| pY766 vs pFRS2 | –0.54±0.17 | –0.83±0.16 |
| pPLCγ vs pFRS2 | –0.37±0.20 | –0.79±0.13 |
| pY653/4 vs downregulation | –0.36±0.26 | –0.36±0.16 |
| pY766 vs downregulation | –0.29±0.27 | –0.41±0.19 |
| pPLCγ vs downregulation | –0.12±0.28 | –0.37±0.16 |
| pFRS2 vs downregulation | 0.25±0.27 | 0.42±0.15 |
| Collagen loss vs growth arrest | –1.77±0.20 | –1.13±0.18 |

whether a response is preferentially engaged by one of the ligands ($\beta \neq 0$) or whether both responses are activated similarly by both ligands ($\beta=0$). We refer to the entire set of coefficients as a 'bias map'.

We assessed statistical significance of the differences in bias coefficients for each pair of responses using ANOVA as described in Materials and methods. The results of the statistical analysis are shown in *Table 2*, where gray shading indicates statistical significance between either FGF4 or FGF9, on one hand, and the reference ligand FGF8, on the other. We see no statistical significance between FGF4 and FGF9 (*Supplementary file 4*).

Based on the ANOVA, we conclude that FGF8 is biased toward phosphorylation of FRS2, against phosphorylation of Y653/654 and Y766, and against PLCγ activation, as compared to FGF9. Furthermore, FGF8 is biased toward phosphorylation of FRS2 and against phosphorylation of Y653/654 and Y766, as compared to FGF4.

## Phosphorylation time courses cannot explain the existence of ligand bias

Previously, it has been suggested that ligand bias may arise due to differences in the time courses of phosphorylation (*Freed et al., 2017*; *Kiyatkin et al., 2020*). We therefore sought to compare the phosphorylation of the activation loop tyrosines Y653/654 and FRS2 over time. We chose these two particular responses because FGF8 is biased toward FRS2 and against Y653/654 FGFR1 phosphorylation, when compared to FGF4 and FGF9 (*Table 2*). Thus, we complemented the Y653/654 phosphorylation time course in *Figure 4A*; *Figure 4—source data 1*, acquired at high concentration (130 nM FGF), with kinetics measurements of Y653/654 phosphorylation at low (2.6 nM) FGF concentration, as well as FRS2 phosphorylation at low (2.6 nM) and high (130 nM ligand) FGF concentrations. Three independent time courses were acquired for each ligand-receptor pair, and the results were averaged. Data, scaled such that the maximal phosphorylation is set to 1, are shown in *Figure 4B–D*.

We observe differences in the time courses of Y653/654 and FRS2 phosphorylation. In the case of FRS2, we always observe fast accumulation of phosphorylated FRS2, which then decreases over time. Such a behavior has been observed for other RTKs and can be explained by the fact that auto-phosphorylation within the RTK dimers/oligomers occurs faster than phosphatase-mediated de-phosphorylation, after which a steady state is established (*Suwanmajo and Krishnan, 2013*; *Lax et al., 2002*). An early maximum in phosphorylation (at 5 min) is seen for FRS2 at both low and high FGF concentration, and is very pronounced for high FGF concentration. An early phosphorylation maximum is also observed for Y653/654 FGFR at high FGF concentration, but not at low FGF concentration.

When comparing the time courses of FGFR1 phosphorylation, in every panel of *Figure 4*; *Figure 4—source data 1*, we do not see discernable differences in the time course of FGF8-induced phosphorylation, as compared to FGF4 or FGF9. Thus, the time course of phosphorylation alone cannot identify FGF8 as a biased ligand, when compared to FGF4 or FGF9.

## Differences in cellular responses and demonstration of functional bias

Biased signaling manifests itself in different cellular responses, which can be cell-specific (*Kenakin, 2019*). We therefore investigated if 293T cells expressing FGFR1 respond differently when stimulated with FGF8 and FGF9. The effects of these two biased FGFs can be directly compared at high concentration as their phosphorylation responses come to a plateau (*Figure 3*). We compared FGFR1 clearance from the plasma membrane due to FGF-induced uptake, cell apoptosis, and cell viability for the 293T cells used in the western blotting experiments. These cellular responses have been previously reported to occur downstream of FGFR1 activation (*Tomlinson et al., 2009*; *Xie et al., 2020*; *Xian et al., 2007*).

To study FGFR1 endocytosis, we measured the decrease in FGFR1-YFP concentration in the plasma membrane in response to a 2 min treatment of 130 nM of FGF8 or FGF9. After fixation, the plasma membranes in contact with the substrate were imaged on a confocal microscope, and the fluorescence intensities for hundreds of cells stimulated with either FGF8, FGF9, or no ligand were recorded. The average fluorescence intensities for the three groups, from three independent experiments, are shown in *Figure 5A*; *Figure 5—source data 1*. We observed a decrease in fluorescence, corresponding to a decrease in plasma membrane concentration of FGFR1-YFP in response to FGF8, as compared to FGF9 and no ligand. This effect was statistically significant by ANOVA ($p < 0.05$). On the other hand, there were no statistically significant differences in FGFR1-YFP membrane concentrations between the FGF9-treated and control groups. These results indicate that FGF8 is more efficient at inducing FGFR1 removal from the plasma membrane immediately after ligand addition, as compared to FGF9. However, the total FGFR1 expression was not affected by a 2 min treatment with ligand (*Figure 5—figure supplement 1*), consistent with the expectation that the FGF8-induced decrease in FGFR1 plasma membrane concentration in *Figure 5A* reflects enhanced internalization that precedes degradation.

We next sought to measure and compare the relative amounts of live FGFR1-expressing 293T cells after stimulation with FGF and after 6 days of starvation, using an MTT assay. The read-out as a function of FGF concentration is shown in *Figure 5B*; *Figure 5—source data 1*, revealing a modest decrease in cell viability with increasing concentration for both FGF8 and FGF9. Data were fitted to linear functions, and the slopes were compared using a t-test. Results show that there is a significant difference in cell viability when cells are stimulated with FGF8, as compared to FGF9 ($p < 0.015$). Further, the effects of the two ligands are significant when compared to the case of no FGF.

To monitor the apoptosis of cells exposed to different FGF8 and FGF9 concentrations, we used a commercial kit that measures the combined activity of caspases 3 and 7 (Magic Red Caspase Kit) through an increase in caspase substrate fluorescence. Experiments were conducted at 0, 20, 40, and 100 nM FGF, under starvation conditions, and the results for each set of experiments were scaled such that caspase activity was 1.0 in the absence of FGF. Data in *Figure 5C*; *Figure 5—source data 1* show the scaled caspase activity as a function of FGF concentration. We see a significant increase in caspase activity in the case of FGF8, but not FGF9. Data were fitted to linear functions, and the best-fit slopes were compared using a t-test. The difference between the slopes was statistically significant ($p < 0.005$), indicating that FGF8 promotes apoptosis more efficiently than FGF9 in 293T cells. We thus observe that FGF8 is more efficient than FGF9 at promoting FGFR1 clearance from the plasma membrane, apoptosis, and cell viability under starvation conditions. This observation is a manifestation of the ligand bias seen upstream.

Since the FGF-induced effects on 293T cell responses were modest, and the dose-response curves did not exhibit rectangular hyperbolic shapes to allow bias coefficient calculations, we quantified two well-known functional responses of RCS cells to FGF treatment: growth arrest and loss of collagen type 2 expression (*Fafilek et al., 2017*; *Krejci et al., 2010*). We performed the experiments with the RCS[Fgfr1] cells used to acquire the data in *Figure 1*. Collagen 2 amounts were measured using western blotting after 48 hr of treatment with FGFs at varying concentrations (*Figure 5D and E*, *Figure 5—source data 4*). Similarly, the growth arrest of RCS[Fgfr1] cells was determined after 72 hr of

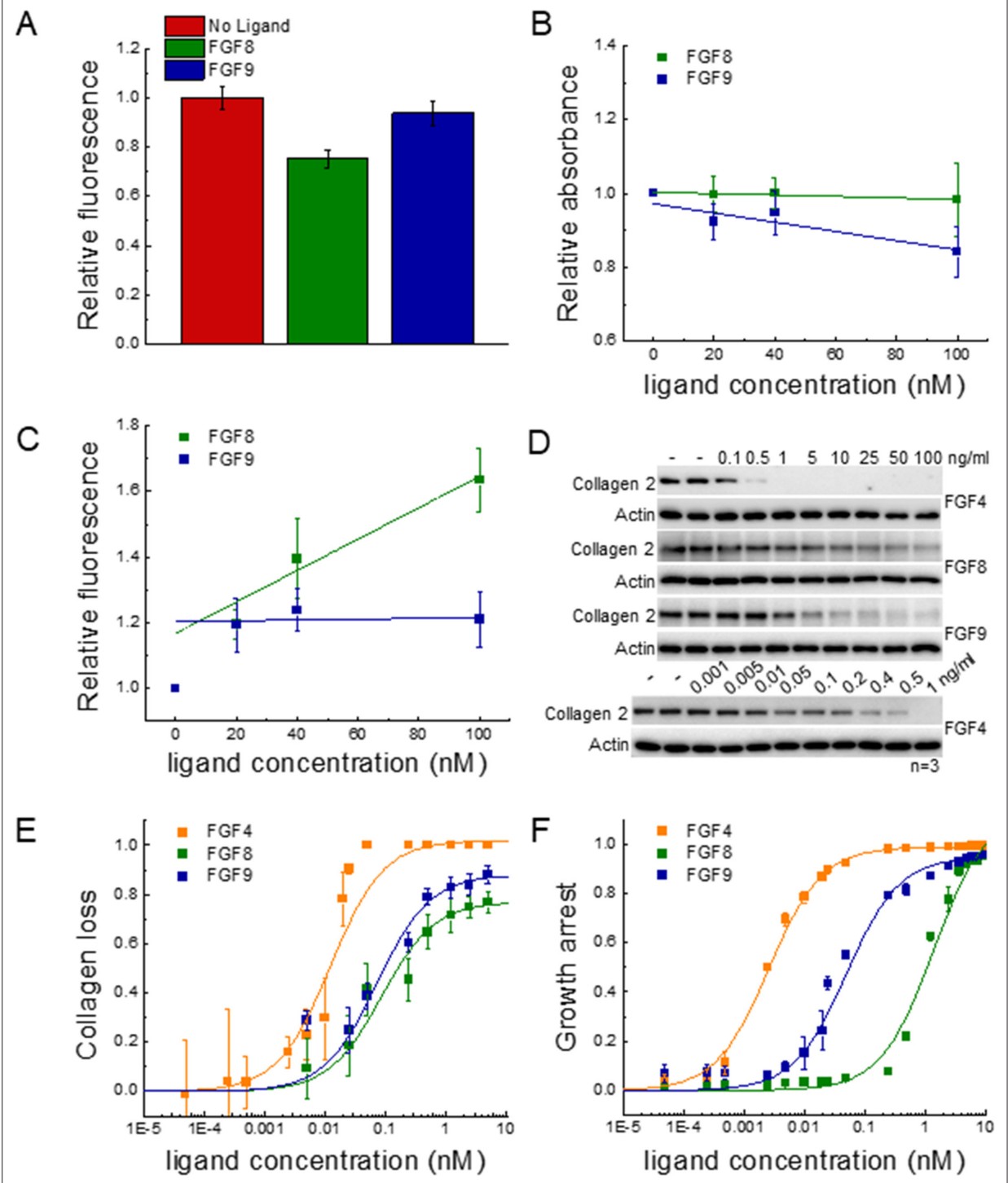

**Figure 5.** Functional FGFR1-mediated responses to different ligands. (**A**) FGFR1 concentration in the plasma membrane of HEK 293T cells at t=2 min following ligand addition for FGF8, FGF9, and no ligand control. (**B**) HEK 293T cell viability after ligand exposure and 6 days of starvation for varying ligand concentrations. (**C**) Apoptosis of HEK 293T cells under starvation conditions, exposed to varying concentrations of FGF8 and FGF9. Results are summarized in *Supplementary file 5*. (**D**) RCS^Fgfr1 cells were treated with FGF4, FGF8, and FGF9 for 48 hr, and the levels of collagen type 2 were determined by western blot. Actin serves as a loading control. (**E**) Dose-response curves describing collagen type 2 loss. (**F**) Dose-response curves for growth arrest of RCS^Fgfr1 cells after 72 hr, in response to FGF4, FGF8, and FGF9. Data are from at least three biologically independent experiments.

The online version of this article includes the following source data and figure supplement(s) for figure 5:

**Source data 1.** FGFR1 membrane concentration, cell viability, and apoptosis data plotted in *Figure 5A, B, and C*.

*Figure 5 continued on next page*

*Figure 5 continued*

**Source data 2.** Original files for the western blots in *Figure 5D*.

**Source data 3.** PDF containing *Figure 5D* and original western blots.

**Source data 4.** Collagen amounts, used to generate the dose-response curves for collagen type 2 loss in *Figure 5E*.

**Source data 5.** Cell counts, used to generate the dose-response curves for growth arrest in *Figure 5F*.

**Figure supplement 1.** Total cellular expression of FGFR1 in the stable FGFR1 line.

**Figure supplement 1—source data 1.** Original files for the western blots in *Figure 5—figure supplement 1*.

**Figure supplement 1—source data 2.** PDF containing *Figure 5—figure supplement 1* and original western blots.

stimulation with the three FGFs (*Figure 5F*, *Figure 5—source data 5*). The dose-response curves in *Figure 5E and F* are distinctly different from each other and all exhibit rectangular hyperbolic shapes. We see that FGF4, FGF8, and FGF9 have very different effects on the two functional responses that we quantified.

The dose-response curves were fitted using *Equation 1* and the $EC_{50}$ and $E_{top}$ values are reported in *Table 1*. The three FGFs exhibit different potencies, with FGF4 being the most potent inducer of both responses. The potencies of FGF8 and FGF9 for collagen 2 reduction are similar, but their potencies to induce growth arrest differ significantly. The efficacies of the three ligands in inducing collagen 2 reduction are different, with FGF4 behaving as full agonist and FGF8 and FGF9 as partial agonists. The efficacies in inducing growth arrest are similar for the three FGFs.

The bias coefficients are reported in *Table 2*. FGF8 is strongly biased toward collagen loss and against growth arrest, when compared to FGF4 and FGF9. The effect is highly significant (*Supplementary file 4*).

## Structural determinants behind FGFR1 biased signaling

We sought possible structural determinants behind the observed FGF bias. For GPCRs, it is now well established that different GPCR ligands stabilize different receptor conformations, where each of the conformations has a preference for a subset of downstream signaling molecules (either G proteins or arrestins) (*Kahsai et al., 2011*; *Liu et al., 2012*; *Kufareva et al., 2017*; *Zheng et al., 2017*; *Wacker et al., 2013*). We therefore asked if structural differences in the FGF4, FGF8, and FGF9-bound FGFR1 dimers may explain bias for FGF8, as compared to FGF4 and FGF9.

It is known that FRS2 binds to the FGFR1 JM domain in a ligand-independent manner (*Gotoh, 2008*; *Ong et al., 2000*). It is also believed that the conformation of the JM domain of RTKs is influenced by the conformation of the TM domain dimer in response to ligand binding (*Doerner et al., 2015*; *Scheck et al., 2012*). Further, the FGFR1 TM domain has already been shown to sense the identity of the bound FGF, either FGF1 or FGF2, and adopt a different TM domain dimer conformation (*Sarabipour and Hristova, 2016*). We therefore sought to investigate the possibility that the TM domains of FGFR1 self-associate differently when FGF8, as compared to FGF4 and FGF9, is bound to the FGFR1 EC domain.

A Förster resonance energy transfer (FRET)-based methodology has been instrumental in revealing differences in FGFR TM domain dimer conformations in response to FGF binding (*Sarabipour and Hristova, 2016*). In these experiments, we monitored differences in the IC distances and relative orientation of fluorescent protein reporters when different ligands bind to the EC domain. We used truncated FGFR1 constructs, which contain the entire EC and TM domains of FGFR1 followed by a flexible linker and either mCherry or YFP. The fluorescent proteins mCherry and YFP are a FRET pair and thus report on the separation between the C-termini of the TM domains in a FGFR1 dimer (*Sarabipour and Hristova, 2016*). Measurements were performed in plasma membrane-derived vesicles, as previously described (*Sarabipour et al., 2015*). High concentration of ligand (250 nM) was used, to ensure that all receptors are ligand-bound. We followed a quantitative protocol (termed QI-FRET) which yields (i) the FRET efficiency, (ii) the donor concentration, and (iii) the acceptor concentration in each vesicle (*King et al., 2016*). A few hundred vesicles, imaged in multiple independent experiments, were analyzed and the data were combined.

*Figure 6A*, left (*Figure 6—source data 1*, *Figure 6—source data 2*, *Figure 6—source data 3*), shows the FRET efficiencies as a function of total FGFR1 concentration in the presence of the three ligands, where each data point corresponds to one individual vesicle derived from one cell. The

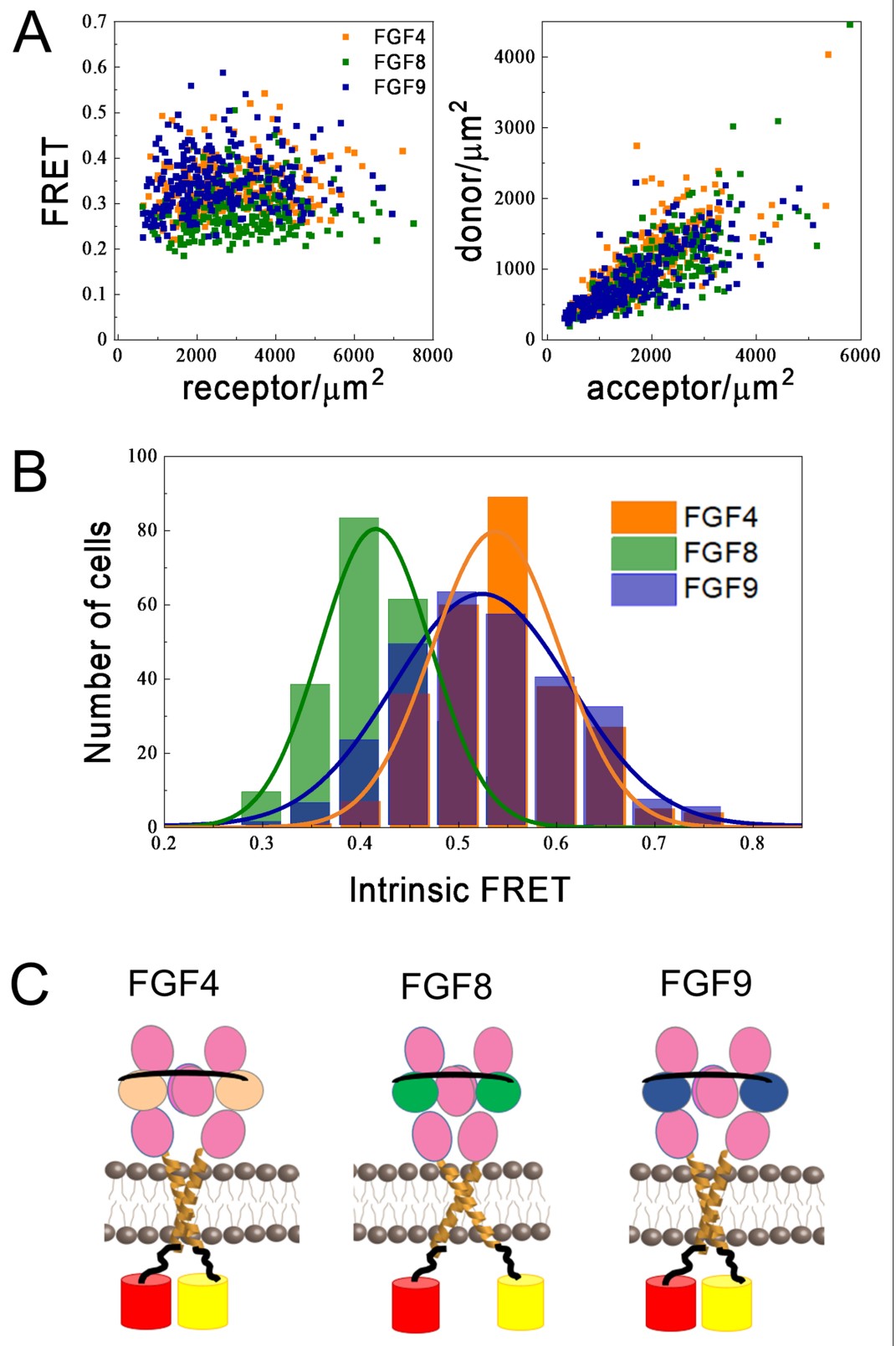

**Figure 6.** Differences in FGFR1 transmembrane domain association in response to FGF ligands. (**A** and **B**) Förster resonance energy transfer (FRET) data for ECTM-FGFR1-YFP and ECTM-FGFR1-mCherry in the presence of saturating FGF4 (orange), FGF8 (green), or FGF9 (blue) concentrations. (**A**) Measured FRET efficiencies versus total receptor (ECTM-FGFR1-YFP + ECTM-FGFR1-mCherry) concentrations and measured donor (ECTM-

*Figure 6 continued on next page*

*Figure 6 continued*

FGFR1-YFP) concentrations versus acceptor (ECTM-FGFR1-YFP) concentrations in single vesicles. (**B**) Histograms of single-vesicle intrinsic FRET values. Intrinsic FRET is a measure of the separation between the fluorescent proteins. Different intrinsic FRET values were measured for FGF8 and FGF4/FGF9. (**C**) Graphical representation of experimental results showing that in the presence of FGF8 the transmembrane (TM) C-termini are positioned further apart from each other, as compared to the cases of FGF4 and FGF9.

The online version of this article includes the following source data for figure 6:

**Source data 1.** FGF4 Förster resonance energy transfer (FRET) data.

**Source data 2.** FGF8 Förster resonance energy transfer (FRET) data.

**Source data 3.** FGF9 Förster resonance energy transfer (FRET) data.

total FGFR1 concentration, on the x axis in *Figure 6A*, left, is the sum of ECTM FGFR1-YFP and ECTM FGFR-mCherry concentrations. In *Figure 6A*, right, we further show the expressions of ECTM FGFR1-YFP and ECTM FGFR1-mCherry in the individual vesicles. Because transient expression levels vary from cell to cell, the vesicles produced in a single transfection experiment had a wide range of receptor concentrations.

We see that the FRET efficiencies in *Figure 6A* do not depend on the concentration, over a broad receptor expression range, indicative of constitutive FGFR1 association in the presence of FGF4, FGF8, and FGF9. In this case, the FRET efficiency depends only on (i) the fraction of acceptor-labeled FGFR1, which is directly calculated from the data in *Figure 6A* and (ii) the so-called intrinsic FRET, a structural parameter which depends on the positioning and dynamics of the fluorophores (*Chen et al., 2010a*). An intrinsic FRET value was calculated for each vesicle from the data in *Figure 6A* using *Equation 5*. The values were binned to generate a histogram and are shown in *Figure 6B*; . These histograms were fitted with Gaussians, yielding intrinsic FRET values of 0.54±0.01, 0.42±0.01, and 0.52±0.01, for the cases of FGF4, FGF8, and FGF9, respectively. Thus, intrinsic FRET is lower for FGF8-bound ECTM FGFR1, as compared to FGF4 and FGF9-bound ECTM FGFR1. Since the fluorescent proteins were attached directly to the TM domains via flexible linkers, the measured differences in intrinsic FRET reflect differences in the separation of the C-termini of the TM domains in the presence of FGF8, compared to FGF4 and FGF9. Thus, the FGFR1 TM domains self-associate in different manner when FGF8 is bound, as compared to the cases when FGF4 or FGF9 are bound.

The effective distances between the fluorescent proteins in the ECTM FGFR1 dimers are calculated using *Equation 6*, assuming random orientation of the fluorophores (justified because they were attached via flexible linkers). In the presence of FGF4 and FGF9, the effective distance between the fluorescent proteins is the same, 55±1 and 56±1 Å ($p>0.05$). In the presence of FGF8, the effective distance between the fluorescent proteins is significantly higher, 60±1 Å ($p<0.01$) as shown in *Figure 6C*. These structural differences may underlie the observed signaling bias in response to FGF8, as compared to FGF4 and FGF9.

## Discussion

The most significant discovery in this work is the existence of bias in FGFR1 signaling in response to three FGF ligands that are encountered during limb development. This concept of 'ligand bias' is a relatively novel concept in RTK research (*Karl et al., 2020*). It describes the ability of ligands to differentially activate signaling pathways (*Kenakin, 2016*; *Gundry et al., 2017*; *Smith et al., 2018*). Ligand bias has been studied primarily in the context of GPCRs and has transformed the fundamental understanding of GPCR signaling (*Gundry et al., 2017*). Importantly, these investigations have produced optimized protocols to identify and quantify bias that are directly applicable to RTKs (*Karl et al., 2020*). Here, we use such quantitative protocols to demonstrate that FGF8 but not FGF4 or FGF9 preferentially induces FRS2 phosphorylation, when compared to the phosphorylation of FGFR1 tyrosines. We also demonstrated that FGF8 preferentially induces collagen 2 loss over growth arrest, as compared to FGF4 and FGF9.

It must be noted that the term 'RTK ligand bias' is often used in the literature to indicate any difference in RTK signaling due to ligands. Here, we use this term in the context of its classical definition in pharmacology, namely the differential activation of at least *two* different signaling responses (*Kenakin, 2016*; *Gundry et al., 2017*; *Smith et al., 2018*). The term 'bias' is distinctly different from

the potencies and efficacies of a ligand for one particular response. For example, in *Figure 5E* we see that FGF8 has both lower potency and lower efficacy than FGF4, but these data cannot provide any information about bias. Bias can be assessed only if the data in *Figure 5E* are compared to the data in *Figure 5F*, leading us to conclude that FGF8 is biased toward collagen 2 loss and against growth arrest, as compared to FGF4, despite the fact that FGF8 has the lowest potency and the lowest efficacy in inducing collagen loss. Potency, efficacy, and bias describe distinct aspects of RTK signaling, yet the differences in these characteristics have not been explicitly considered in prior studies of FGF signaling.

Ligand bias studies require the comparison of at least two responses and at least two ligands (*Kenakin, 2016*; *Gundry et al., 2017*; *Smith et al., 2018*). Prior studies of the effects of different FGFs on FGFR signaling have utilized BAF/3 cells (*Ornitz et al., 1996*; *Zhang et al., 2006*; *Buchtova et al., 2015*). In these cells, only a single response, cell proliferation, can be quantified and compared, and thus these cells cannot be used for bias studies. In HEK 293T cells, the functional effects due to FGF4, FGF8, and FGF9 are modest (*Figure 5A–C*). However, here we were able to measure dose-response curves for two functional responses in engineered RCS^Fgfr1 cells (*Figure 5E and F*). These cells allow us to model processes occurring in the developing mammalian limb, where the three FGF ligands (FGF4, FGF8, FGF9) released by the ectoderm at the surface of the limb bud signal to the underlying mesenchymal cell which expresses just one FGF receptor, FGFR1c (*Mariani and Martin, 2003*; *Tabin and Wolpert, 2007*). In RCS cells, cellular phenotypes caused by FGF signaling can be quantified, and thus RCS cells have been used in studies exploring the mechanisms of FGF-FGFR signaling, including mechanisms behind FGF regulation of the cell cycle, cell proliferation, differentiation, premature senescence, loss of EC matrix, interplay between FGF and WNT signaling, cytokine and natriuretic peptide signaling, and others (*Fafilek et al., 2017*; *Krejci et al., 2010*; *Raucci et al., 2004*; *Priore et al., 2006*; *Kamemura et al., 2017*; *Kolupaeva et al., 2013*; *Dailey et al., 2003*; *Rozenblatt-Rosen et al., 2002*). In addition, treatments inhibiting pathological FGFR signaling, which are now either in human trials (RBM007, meclozine) or FDA-approved (vosoritide), were initially developed in RCS cells, benefiting from the well-characterized molecular mechanisms of FGF signaling in these cells (*Kimura et al., 2021*; *Wendt et al., 2015*; *Matsushita et al., 2013*). Here, we show that RCS cells can also be used to identify biased FGF ligands.

Thus far, differential effects in the signaling of one FGFR in response to different FGF ligands have been attributed to differences in ligand binding. It can be reasoned, however, that differences in ligand binding strengths, alone, cannot explain differential signaling. Indeed, if the differences between ligands are only in the binding strength, then a strongly binding ligand, at low concentration, will act identically to weakly binding ligand at high concentration. Here, we discovered, using tools that are novel for the RTK field, that there are also qualitative differences in the actions of the ligands. FGF8 preferentially activates some of the probed downstream responses (FRS2 phosphorylation and collagen loss), while FGF4 and FGF9 preferentially activate different probed responses (FGFR1 phosphorylation and growth arrest). These effects occur in addition to previously measured differences in ligand binding coefficients (*Mohammadi et al., 2005*).

Having established the existence of bias in FGFR1 signaling, we sought the mechanism that allows for the recognition of FGF8 binding to the FGFR1 EC domain, as compared to FGF4 and FGF9. We measured FRET efficiency when the fluorescent proteins were attached to the C-termini of the TM helices via flexible linkers. We showed that the FRET efficiency is different when FGF8 is bound to the EC domain, as compared to the cases of bound FGF4 and FGF9. These results suggest that the C-termini of the TM helices are spaced further apart when FGF8 is bound, as compared to FGF4 and FGF9. Thus, the FGFR1 TM domains must sense the identity of the ligand that is bound to the EC domain. FRS2 binds to the JM domain, which immediately follows the TM domain (*Gotoh, 2008*; *Ong et al., 2000*; *Hadari et al., 1998*). It is easy to envision that the conformation of the TM domain affects FRS2 behavior, as the FRS2 binding site is just 32 amino acids from the TM domain C-terminus. The larger TM domain separation in the FGF8-bound FGFR1 dimers may facilitate the preferential phosphorylation of FRS2 over other sites. Alternatively, the smaller TM domain separation upon binding of FGF4 and FGF9 may disfavor FRS2 phosphorylation. Further, the differential phosphorylation of FRS2 may be due to a different mode of FRS2 binding to FGFR1, differences in the accessibility of FRS2 by the active site of the FGFR1 kinase, or different accessibility of FRS2 by phosphatases.

The experimental data in *Figure 6* hint at the possibility that ligand bias arises due to differences in FGFR1 dimer conformations. If this is so, then conformational differences in the signaling complex in the plasma membrane may underlie biased signaling for both RTKs and GPCRs, the two largest receptor families in the human genome. It is important to note, however, that there are researchers who disagree that structural information can be propagated along the length of the RTK, because the linkers between the RTK domains are flexible (*Freed et al., 2017*; *Lu et al., 2010*; *Bocharov et al., 2017*). If so, how do the kinases sense which ligand is bound to the EC domain? Some have proposed that differential downstream signaling occurs as a result of different kinetics of receptor phosphorylation/de-phosphorylation in response to different ligands (*Freed et al., 2017*). There is a report that tyrosine phosphorylation of EGFR is more sustained in response to epigen and epiregulin than to EGF (*Freed et al., 2017*). Specifically, it was found that EGF-induced EGFR phosphorylation of Y845, Y1086, and Y1173 returned to baseline much faster, as compared to phosphorylation in response to epigen and epiregulin (*Freed et al., 2017*). The authors argued that differences in kinetics do not depend on the specific ligand concentration used, and they attributed the characteristic kinetics of the response to the identity of the bound ligand. In studies with other RTKs, however, the concentration of the ligand strongly influenced the kinetics of the response, questioning the applicability of this model to all RTKs (*Wykosky et al., 2008*; *Gomez-Soler et al., 2019*). Here, we show that FGFR1 kinetics of phosphorylation also depend strongly on FGF concentration. Furthermore, we do not observe a discernable correlation between kinetics of FGFR1 phosphorylation and ligand bias.

In this study, we observe marked differences in the potencies of the three FGFs. Indeed, 50% of maximum phosphorylation in 293T cells is reached for ~0.5 nM FGF4, ~2 nM FGF9, and ~10 nM FGF8. Thus, the same level of phosphorylation of a specific tyrosine can be achieved for much lower concentrations of FGF4, as compared to FGF8 or FGF9. The potencies of FGF4 in inducing functional responses are also the highest in RCS cells. We also observe differences in the efficacies of the three FGF ligands. The highest phosphorylation in *Figure 3* is achieved in response to high concentrations of FGF8. Thus, FGF8 is a full agonist while FGF4 and FGF9 are partial agonists for most of the responses (with the exception of FRS2 phosphorylation). Along with differences in potencies and efficacies, we observe a significant difference in the very shape of the dose-response curves. In particular, the FGF4 dose response does not follow the anticipated rectangular hyperbolic shape, unlike the cases of FGF8 and FGF9. We see that the FGF4-induced phosphorylation of FGFR1 and the effector molecules increases up to ~3 nM FGF4, and then gradually decreases as FGF4 concentration is further increased. The mechanism behind the decrease in activation at high FGF4 concentration is unknown, but our data suggest that it cannot be explained with differential kinetics of de-phosphorylation/downregulation. Furthermore, there is no correlation between the unusual shape of the FGF4 dose response and ligand bias, as FGF4 is not biased when compared to FGF9. Instead, we see an intriguing correlation between the decrease in FGFR1 phosphorylation and the appearance of FGFR1 oligomers that are larger than dimers. Indeed, the FIF data in *Figure 2* suggests that high FGF4 concentration promotes oligomerization of FGFR1. Oligomers are observed in full-length FGFR when bound to FGF4, but not when the kinase domains are deleted (based on FIF measurements). This finding suggests that, at least in part, the oligomers are stabilized by kinase-kinase contacts.

We can explain the shape of the FGF4 dose-response curves if we assume that phosphorylation is less efficient in FGFR1 oligomers as compared to FGFR1 dimers. It is important to note that EGFR has also been proposed to form oligomers in response to EGF, in addition to dimers (*Needham et al., 2016*; *Huang et al., 2016*). EGFR dimers and oligomers have been proposed to have different activities (*Needham et al., 2016*; *Huang et al., 2016*). It has been suggested that in the EGFR oligomer, each EGFR kinase may be able to phosphorylate and be phosphorylated by multiple neighboring kinases, leading to higher overall activation. Such a view, however, is not consistent with the FGFR1 data presented here. Perhaps, the functional role of FGFR1 oligomers is very different from that of the EGFR oligomers, as oligomerization seems to inhibit FGFR1 signaling.

In conclusion, we initiated this study in search of a possible difference in the response of FGFR1 to three FGF ligands. We found not one, but many differences. We observed quantitative differences in most aspects of the FGFR1 signaling response: ligand-induced oligomerization, potency, efficacy, and conformations of FGFR1 TM domain dimers. We also discovered the existence of ligand bias in FGFR1 signaling. The quantitative tools that we used were instrumental in revealing these differences, based on the analysis of FGFR1 dose-response curves for the three ligands. Of note, such dose-response

curves are typically collected for GPCRs (*Kenakin, 2019*; *Luttrell, 2014*; *Correll and McKittrick, 2014*; *Kenakin, 2011*), but rarely for RTKs. Future use of such quantitative tools for other RTKs may similarly reveal unexpected differences in response to different ligands and may uncover exciting new biology.

All 58 RTKs have been implicated in many growth disorders and cancers (*Lemmon and Schlessinger, 2010*; *Robertson et al., 2000*; *Browne et al., 2009*; *Li and Hristova, 2006*), and have been recognized as important drug targets. Inhibitors that specifically target the RTKs have been under development for decades now. Recent years have seen significant improvements in RTK-targeted molecular therapies, but these therapies have not yet fundamentally altered the survival and the quality of life of patients (*Le et al., 2021*; *Kumar et al., 2020*). This limited progress may be because RTK signaling is much more complex than currently appreciated, as demonstrated here for FGFR1. Furthermore, it is possible that ligand bias plays an important role in RTK-linked pathologies. The practice of evaluating bias for novel RTK-targeting therapeutics may empower the discovery of a new generation of biased RTK inhibitors that selectively target pathogenic signaling pathways, and thus exhibit high specificity and low toxicity.

# Materials and methods

**Key resources table**

| Reagent type (species) or resource | Designation | Source or reference | Identifiers | Additional information |
|---|---|---|---|---|
| Antibody | Fgfr1 (rabbit monoclonal) | Cell Signaling | 9740 | (1:1000) |
| Antibody | Fgfr2 (rabbit polyclonal) | Santa Cruz | sc122 | (1:1000) |
| Antibody | Collagen 2 (rabbit polyclonal) | Cedarlane | CL50241AP | (1:1000) |
| Antibody | Actin (mouse monoclonal) | Cell Signaling | 3700 | (1:1000) |
| Antibody | Vinculin (rabbit monoclonal) | Cell Signaling | 13901 | (1:1000) |
| Antibody | pERK (rabbit polyclonal) | Cell Signaling | 9101 | (1:1000) |
| Antibody | Anti-Y653/654 (rabbit polyclonal) | Cell Signaling | 3471S | (1:1000) |
| Antibody | Anti-pY766 FGFR1 (rabbit monoclonal) | Cell Signaling | 2544S | (1:1000) |
| Antibody | Anti-pFRS2 (rabbit monoclonal) | Cell Signaling | 3861S | (1:1000) |
| Antibody | Anti-pPLCγ (rabbit polyclonal) | Cell Signaling | 2821S | (1:1000) |
| Antibody | Anti-PLCγ (rabbit polyclonal) | Cell Signaling | 2822S | (1:1000) |
| Antibody | Anti-V5 (mouse monocolonal) | Invitrogen | 46-0705 | (1:1000) |
| Antibody (secondary) | Anti-rabbit | Promega | W4011 | (1:10,000) |
| Antibody (secondary) | Anti-mouse | Millipore Sigma | A6782 | (1:10,000) |
| Cell line (*Homo sapiens*) | HEK 293T FGFR1 | This paper | | Stable cell line developed and maintained by Hristova lab, identity authenticated by SRT profiling, negative for mycoplasma |
| Cell line (*Rattus norvegicus*) | RCS WT | PMID:749928 | | A gift from Benoit de Crombrugghe |
| Cell line (*Rattus norvegicus*) | RCS Fgfr1-4 null | PMID:33952673 | | |
| Cell line (*Rattus norvegicus*) | RCS^Fgfr1 | This paper | | Progenitors: Fgfr3/4 KO RCS cells from Carmine Settembre, identity authenticated by SRT profiling, negative for mycoplasma |

*Continued on next page*

*Continued*

| Reagent type (species) or resource | Designation | Source or reference | Identifiers | Additional information |
|---|---|---|---|---|
| Chemical compound, drug | 2× Laemmli Buffer | Bio-Rad | 1610737 | |
| Chemical compound, drug | Tris/Gly/SDS running buffer | Bio-Rad | 1610732 | |
| Chemical compound, drug | Transfer buffer | Bio-Rad | 1610734 | |
| Chemical compound, drug | Fugene HD | Promega | E2311 | |
| Commercial assay or kit | Bio-Rad Mini-Protean TGX precast gels | Bio-Rad | 4561026 | |
| Commercial assay or kit | PVDF membranes | Bio-Rad | 1620177 | |
| Commercial assay or kit | West Femto Supersignal | Thermo Fisher Scientific | 1706435 | |
| Commercial assay or kit | Bio-Rad Magic Red Caspase 3-7 kit | Bio-Rad | ICT 935 | |
| Commercial assay or kit | MTT Cell Proliferation Assay Kit | Cell BioLabs | CBS-252 | |
| Other | Iblot 2 Gel Transfer Device | Thermo Fisher Scientific | IB21001 | Equipment for transfer |
| Gene (*Rattus norvegicus*) | *Fgfr1* | Ensembl | Ensembl:ENSRNOG00000016050 | |
| Peptide, recombinant protein | FGF4 | R&D Systems | 235-F4-025 | |
| Peptide, recombinant protein | FGF8 | R&D Systems | 423-F8-025 | |
| Peptide, recombinant Protein | FGF9 | R&D Systems | 273-F9-025 | |
| Recombinant DNA reagent | FGFR1-ECTM-eYFP (plasmid) | PMID:26725515 | | YFP-Dr M. Betenbaugh, FGFR1 in pRK5- Dr M Mohammadi, into pcDNA3.1 vector |
| Recombinant DNA reagent | FGFR1-ECTM-mCherry (plasmid) | PMID:26725515 | | pRSET-mCherry- Dr R.Tsien, FGFR1 in pRK5- Dr M Mohammadi, into pcDNA3.1 vector |
| Recombinant DNA reagent | pKrox(MapERK)d1EGFP | This paper | Addgene plasmid #214912 | Progenitors: pKrox24(MapErk) DsRed (PMID:28199182), pTR01F (PMID:24376882), d1EGFP (PMID:16508309), PCR of mEgr1 3'UTR |
| Software | FIF software | PMID:31110281 | | |
| Software | Mathematica | Wolfram | 13 | |
| Software | Prism | GraphPad | 9.2.0 | |

## Cell culture

Human embryonic kidney cells (HEK 293T), stably transfected with FGFR1-YFP, and RCS$^{Fgfr1}$ cells were grown in Dulbecco's modified eagle medium (DMEM; Thermo Fisher, PN 31600034) with 10% fetal bovine serum supplemented with 3.5 g of glucose and 1.5 g of sodium bicarbonate at 37°C and 5% $CO_2$. Chinese hamster ovarian (CHO) cells were grown in DMEM with 10% fetal bovine serum

supplemented with 0.8 g of glucose, 1.5 g of sodium bicarbonate, and 1% NEAA at 37°C and 5% $CO_2$. The RCS Fgfr1–4 *null* cells were prepared by Crispr/Cas9-mediated inactivation of *Fgfr1*, *Fgfr2*, *Fgfr3*, and *Fgfr4* loci (**Kimura et al., 2021**). RCS$^{Fgfr1}$ cells were prepared by Crispr/Cas9-mediated inactivation of *Fgfr2* in *Fgfr3/4* double knock-out RCS cells (a gift from Carmine Settembre).

## FIF spectrometry

HEK 293T cells, stably transfected with FGFR1-YFP, were seeded on collagen-coated, glass-bottom Petri dishes (MatTek, P35GCOL-1.5-14-C) and allowed to grow to ~70% confluency at 37°C and 5% $CO_2$. Cells were grown in DMEM with 10% fetal bovine serum, supplemented with 3.5 g of glucose and 1.5 g of sodium bicarbonate. Cells were rinsed with 70% swelling media (1:9 serum-free media, di$H_2O$, 25 mM HEPES), and then swelled in 70% swelling media plus ligand for ~5 min before imaging. This treatment minimizes the ruffles in the plasma membrane and prevents endocytosis (**Rauch and Farge, 2000**; **Sinha et al., 2011**).

The plasma membranes facing the support were imaged on a TCS SP8 confocal microscope using a photon counting detector. Images were analyzed using the FIF software (**Stoneman et al., 2019**). Briefly, the membrane was divided into 15×15 pixel regions of interest and the molecular brightness, ε, of each region was calculated as:

$$\varepsilon = \frac{\sigma^2}{\langle I \rangle} - 1$$

where $\langle I \rangle$ is the center of the Gaussian distribution and $\sigma^2$ is the variance in each segment. The brightness values from thousands of regions of interest were binned and histogrammed. The histograms were then normalized to 1.

Since the molecular brightness distribution is log-normal, the values of log(brightness) were histogrammed and fit to a Gaussian function:

$$normalized\ counts = a * e^{\dfrac{-(\theta - m)^2}{2s^2}}$$

Here, θ represents the log of the brightness, m is the mean of the Gaussian, s is the standard deviation of the Gaussian, and a is a constant. The best-fit Gaussian parameters are shown in **Supplementary file 1**.

In order to determine whether the distributions were the same or different, a Z-statistics analysis was used, where the Z value is given by:

$$Z = \frac{m_1 - m_2}{\sqrt{q_1^2 + q_2^2}}$$

Here, $m_1$ and $m_2$ represent the two means of the Gaussians being compared, and $q_1$ and $q_2$ are the standard deviations for each Gaussian divided by the square root of the number of cells analyzed. A minimum of 100 cells were analyzed for each data set.

A Z value of less than 2 means that the data sets are within 2 standard deviations of the mean and are therefore the same, while a Z value greater than 2 means that the two data sets are different (**Anderson et al., 2001**). Calculated Z values can be found in **Supplementary file 2**.

## Western blots

HEK 293T cells stably expressing FGFR1 were seeded in equal volumes into 12-well plates and allowed to grow to ~70–90% confluency, while changing the media after 48 hr. The full media was aspirated and replaced with DMEM without fetal bovine serum. Varying amounts of ligands (R&D Systems; FGF4, #235-F4-025; FGF8, #423-F8-025, and FGF9, # 273-F9-025) were added to each well to create a ligand concentration gradient. Cells were incubated with the ligands at 37°C and 5% $CO_2$ for 20 min and then immediately placed on ice. Media was aspirated and cells were immediately lysed with 2× Laemmli Buffer (Bio-Rad 1610737)+5% BME. Lysates were moved to clean Eppendorf tubes and vortexed for 10 s intervals six times over the course of 5–10 min, staying on ice in the interim. Lysates were centrifuged, boiled at 98°C for 10 min, and then immediately placed on ice until cool. Lysates were then centrifuged again and stored at –20°C for later use. Lysates were thawed on ice and

vortexed immediately prior to loading onto gel. Gels were run on ice at 140 V for 3 hr using Bio-Rad Mini-Protean TGX precast gels in 1× Tris/Gly/SDS running buffer (Bio-Rad #1610732). Gels were then equilibrated with transfer buffer (Bio-Rad #1610734) supplemented with 20% methanol for 10 min. Transfer was performed using the Iblot 2 Gel Transfer Device (Thermo Fisher #IB21001) and nitrocellulose packs (Thermo Fisher IB23001). Nitrocellulose membranes were removed and replaced with PVDF membranes (Bio-Rad #1620177) that had been activated in 100% methanol, all other steps were as prescribed by Thermo Fisher. Transfers were done at 25 V for 7 min. Following transfer, membranes were immediately trimmed and placed in either 5% non-fat milk or 5% BSA in 1× TBS supplemented with 1% Tween (Sigma #P1379) (TBST) depending on the primary antibody used. Blocking was accomplished on a rocker for 1–2 hr at room temperature. Membranes were rinsed 2× with TBST and then primary antibody was added at a 1:1000 dilution. Primary antibodies (Cell Signaling: anti-Y653/654 #3471S, anti-FGFR1 #9740S, anti-pY766 FGFR1, #2544, anti-Actin #3700, anti-pERK1/2 #9101, anti-pFRS2 #3861S, anti-pPLCγ #2821S, anti-PLCγ #2822S, anti-Vinculin #13901; Santa Cruz: anti-FGFR2 #sc122; Cedarlane: anti-collagen 2 #CL50241AP; Invitrogen: anti-V5 #46-0705) were incubated with the membrane overnight at 4°C on a rocker. The primary antibody was removed and the membrane was washed with TBST three times for 5–15 min. The secondary antibody (Promega: anti-rabbit #W4011; Sigma: anti-mouse #A6782) was added at a dilution of 1:10,000 and allowed to incubate on a rocker at room temperature for 1–2 hr. The secondary antibody was removed and the membrane was washed with TBST 3× for 5–15 min. The membrane was incubated with chemiluminescent solution (Thermo Fisher Scientific West Femto Supersignal, #1706435) and imaged on a Bio-Rad Gel-Doc XRS+.

Blot images were stored digitally. The intensity of each band was quantified using ImageJ.

## Scaling of phosphorylation western blot data

To identify and quantify bias, the phosphorylation dose-response curves for the three ligands have to be globally scaled such that the efficacy of the full agonist is set to 1, and the efficacies of the partial agonist are scaled accordingly. Below, we present the protocol. Steps 1 through 3 are used in the literature to scale the individual dose-response curves. Steps 4 through 6 implement the global scaling.

1. Collect at least three independent dose-response curves for each ligand. Each dose response is acquired using one gel.
2. Scale each dose-response curve so max is 1. In other words, set the band intensity for the sample with the highest band intensity to 1.
3. Average the dose-response curves, and reset the maximum value to 1 if needed.
4. For each ligand, re-run three of the samples with intensity 1 on a gel. This is the so-called 'gluing gel'. This gel includes three samples for each ligand, so nine total (see *Figure 3B*).
5. Average the three intensities for each ligand. Identify the ligand that yielded the highest average intensity. Rescale the three averages, such that this highest intensity is set to 1.
6. Multiply the averaged dose-response curves by the respective global scaling coefficients to obtain the globally scaled dose-response curves.

## Scaling of FGFR1 and collagen abundance gels

In this case, the band intensities at zero ligand are the highest and they are set to 1; the rest are scaled accordingly. Results from at least three independent samples are averaged. Then, 1 is subtracted from all the data and the sign is inverted to obtain loss of abundance dose-response curves.

## Fitting of dose-response curves

The scaled dose-response curves were fitted to a rectangular hyperbolic (Hill equation with n=1) given by:

$$response = \frac{[x] * E_{top}}{[x] + EC_{50}} \tag{1}$$

Here, [x] represents the concentration of ligand, while $E_{top}$ corresponds to the plateau at high ligand concentrations and $EC_{50}$ corresponds to the ligand concentration value at which 50% of $E_{top}$ is achieved.

To fit the data to *Equation 1*, we truncated each dose-response curve at the highest ligand concentration that is within 10% of the maximum y value. This took into account that the average y value error is between 5% and 10%. The data was fit in Mathematica with the NonlinearModelFit function using the Levenberg-Marquardt minimization method, while allowing for a maximum of 100,000 iterations. The errors of the fit were weighted as the inverse of the square of the error.

## Calculation of bias coefficients

To calculate bias coefficients, FGF8 was chosen as the reference ligand. Bias coefficients, β, for FGF4 and FGF9 were calculated using *Equation 2*.

$$\beta_{4,9} = Log \left( \left( \frac{E_{top,1}}{EC_{50,1}} * \frac{EC_{50,2}}{E_{top,2}} \right)_{4,9} * \left( \frac{EC_{50,1}}{E_{top,1}} * \frac{E_{top,2}}{EC_{50.2}} \right)_{8} \right) \tag{2}$$

Standard errors of β were calculated using the standard errors of $EC_{50}$ and $E_{top}$, as determined from the fit in *Equation 1*, using the functional approach for multi-variable functions (*Hughes and Hase, 2010*).

To compare bias coefficients for the three ligands and determine statistical significance in a three-way comparison, we follow the protocol in *Karl et al., 2020*, re-writing *Equation 2* as:

$$\beta_{4,9} = \beta'_{4,9} - \beta'_{8} \tag{3}$$

where β' is calculated in *Equation 4*.

$$\beta'_{4,8,9} = Log \left( \frac{E_{top,1}}{EC_{50,1}} * \frac{EC_{50,2}}{E_{top,2}} \right)_{4,8,9} \tag{4}$$

Standard errors of β' were calculated using the standard errors of $EC_{50}$ and $E_{top}$ using the functional approach for multi-variable functions (*Hughes and Hase, 2010*). The β' values and their corresponding errors are reported in *Supplementary file 3*.

Statistical significance of the differences between β' values were calculated with a one-way ANOVA using the multi-variable analysis option in Prism. The data that were inputted were the mean, SEM, and n, the number of points contributing to the fit. In *Figure 3*, n=9 for FGF4, n=10 for FGF8, and n=7 for FGF9. The calculated p-values are shown in *Supplementary file 4*.

## FRET measurements

CHO cells were seeded into tissue culture-treated six-well plates at a density of $2*10^4$ cells/well. 24 hr later, the cells were co-transfected with FGFR1-ECTM-eYFP and FGFR1-ECTM-mCherry using Fugene HD (Promega # E2311) according to the manufacturer's instructions. ECTM constructs included the entire EC domain and TM domain up to residue 402, followed by a $(GGS)_5$ flexible linker and the fluorophore (*Sarabipour and Hristova, 2016*). The amount of DNA as well as the donor to acceptor ratio of the DNA added was varied to achieve a wide range of donor and acceptor expressions (*Sarabipour et al., 2015*).

Cells were vesiculated ~24 hr after transfection as described previously (*Del Piccolo et al., 2012*). Briefly, the cells were rinsed 2 × with 30% PBS and then incubated for 13 hr at 37°C and 5% $CO_2$ in a chloride salt buffer. Vesicles were then transferred to four-well glass-bottomed chamber slides for imaging on a Nikon confocal microscope with a 60× objective to capture images of the equatorial cross-sections of the vesicles in three channels: donor (eYFP), acceptor (mCherry), and FRET (*Sarabipour et al., 2016*). FRET was measured following the QI-FRET protocol (*Sarabipour et al., 2015*; *Chen et al., 2010a*), which yields the donor concentration, the acceptor concentration, and the FRET efficiency in each vesicle. The microscope was calibrated using solutions of fluorescent protein of known concentration, so that the fluorescence intensity could be directly correlated to fluorophore concentration.

In the case of constitutive dimers at high ligand concentration, when FRET does not depend on receptor concentration, the measured FRET and the intrinsic FRET are connected as follows *King et al., 2016*:

$$\tilde{E} = \frac{E}{x_A} \tag{5}$$

Here, $x_A$ is the acceptor fraction, which is measured in each vesicle, along with the FRET efficiency, E. *Equation 5* allows us to directly determine the value of the intrinsic FRET, $\tilde{E}$, in each vesicle. The dependence of the intrinsic FRET, $\tilde{E}$, on the distance between the fluorescent proteins in the dimer, d, is given by *Equation 6*.

$$\tilde{E} = \frac{1}{1 + \left(\dfrac{d}{R_O}\right)^6} \tag{6}$$

Here, $R_o$ is the Förster radius of the FRET pair. For eYFP and mCherry, $R_0$ is 53.1 Å (*Chen et al., 2010a*; *Chen et al., 2010b*). This equation assumes free rotation of the fluorescent proteins. This assumption can be justified because the fluorescent proteins are attached via flexible linkers.

## Ligand-induced FGFR1 removal from the plasma membrane

FGFR1 downregulation in the plasma membrane was assayed by measuring the FGFR1 membrane concentration before and 2 min after ligand addition. FGFR1-YFP in the plasma membrane in contact with the substrate was imaged in a TCS SP8 confocal microscope, and FGFR1-YFP fluorescence per unit membrane area was quantified. HEK 293T cells, stably transfected with FGFR1-YFP, were seeded on collagen-coated, glass-bottom Petri dishes (MatTek, P35GCOL-1.5-14-C) and allowed to grow to ~70% confluency at 37°C and 5% $CO_2$. Cells were rinsed with serum-free media and exposed to 130 nM of either FGF8 or FGF9 or no ligand and incubated at 37°C and 5% $CO_2$ for 2 min. The cells were fixed in a solution of 4% formaldehyde in PBS for 20 min at room temperature. Samples were stored at 4°C prior to imaging. A minimum of 100 cells per condition were imaged, in three independent experiments. The receptor concentration in the membrane of each cell was quantified using the FIF software (*Stoneman et al., 2019*), and results for all analyzed cells per condition were averaged.

## Viability assays

Cell viability was monitored using a MTT Cell Proliferation Assay Kit (Cell BioLabs, #CBS-252). HEK 293T cells, stably transfected with FGFR1-YFP, were seeded in 96-well plates and allowed to grow to ~70% confluency. Media was aspirated and replaced with serum-free media and with varying concentrations of ligand. After ligand addition, cells were kept at 37°C and 5% $CO_2$ for 6 days. Viability was measured according to the manufacturer's protocol. Briefly, CytoSelect MTT Cell Proliferation Assay Reagent was added directly to the cell media. After 3 hr of incubation at 37°C and 5% $CO_2$, the cells were lysed with detergent and kept at room temperature for 2 hr. The absorbance was measured on a Synergy H4 plate reader at 555 nm.

## Apoptosis assays

Apoptosis was probed using the Bio-Rad Magic Red Caspase 3-7 kit (Bio-Rad, ICT 935). HEK 293T cells stably expressing FGFR1-YFP were seeded in 96-well plates and allowed to grow to ~70% confluency. Media was aspirated and replaced with serum-free media with the Magic Red staining solution and varying concentrations of ligand. After ligand treatment the cells were placed in the incubator at 37°C and 5% $CO_2$ for 6 days. The fluorescence of the cleaved substrate (Cresyl Violet) was measured on a Synergy H4 plate reader. The excitation wavelength was 592 nm. Emission was measured at 628 nm.

## ERK activation, cell count, and collagen expression experiments in RCS[Fgfr1] cells

The pKrox(MapERK)d1EGFP reporter was stably expressed in RCS[Fgfr1] cells using the piggyBac transposase. Activation of the pKrox(MapERK)d1EGFP reporter in RCS[Fgfr1] cells was measured using an automated incubation microscope BioStation CT (Nikon, Tokyo, Japan). Phase contrast and fluorescence signal images were automatically acquired every 1 hr during a 48 hr time period. Fluorescence of the reporter was then processed and analyzed with the Nikon BioStation CT software.

For the growth arrest assay, the RCS cells were treated with 1 μg/ml heparin and FGF ligands for 3 days. Cell numbers were determined by counting (Beckman-Coulter). For collagen type 2 expression, the RCS cells were treated with 1 μg/ml heparin and FGF ligands for 48 hr. Collagen type 2 expression was measured by western blotting.

## Materials availability statement

Cell lines created for this work are available upon request.

## Acknowledgements

We thank Marie Tesarova for excellent technical assistance and Carmine Settembre, Telethon Institute of Genetics and Medicine, Italy, for providing *Fgf3/4* double knock-out RCS cells. The work was supported by NIH GM068619 and NSF MCB 2106031 to KH. PK is supported by the Czech Science Foundation (GA19-20123S, GF21-26400K); Ministry of Education, Youth and Sports of the Czech Republic (INTER-ACTION LUAUS23295); Grant Agency of Masaryk University (MUNI/G/1771/2020); National Institute for Cancer Research (Programme EXCELES, ID Project No. LX22NPO5102) - Funded by the European Union - Next Generation EU. BF is supported by the Agency for Healthcare Research of the Czech Republic (NU21-06-00512).

## Additional information

### Funding

| Funder | Grant reference number | Author |
|---|---|---|
| National Institute of General Medical Sciences | GM068619 | Kalina Hristova |
| National Science Foundation | 2106031 | Kalina Hristova |
| Czech Science Foundation | GA19-20123S | Pavel Krejci |
| Czech Science Foundation | GF21-26400K | Pavel Krejci |
| Ministry of Education, Youth and Sports of the Czech Republic | INTER-ACTION LUAUS23295 | Pavel Krejci |
| Grant Agency of Masaryk University | MUNI/G/1771/2020 | Pavel Krejci |
| National Institute for Cancer Research | LX22NPO5102 | Pavel Krejci |
| Agency for Healthcare Research of the Czech Republic | NU21-06-00512 | Bohumil Fafilek |

The funders had no role in study design, data collection and interpretation, or the decision to submit the work for publication.

### Author contributions

Kelly Karl, Conceptualization, Formal analysis, Investigation, Writing - original draft; Nuala Del Piccolo, Tanaya Roy, Investigation, Writing - review and editing; Taylor Light, Pooja Dudeja, Vlad-Constantin Ursachi, Investigation; Bohumil Fafilek, Conceptualization, Investigation, Writing - review and editing; Pavel Krejci, Conceptualization, Supervision, Funding acquisition, Project administration, Writing - review and editing; Kalina Hristova, Conceptualization, Funding acquisition, Project administration, Writing - review and editing

### Author ORCIDs

Nuala Del Piccolo (iD) http://orcid.org/0000-0002-5104-7322
Tanaya Roy (iD) http://orcid.org/0000-0003-0939-7158
Bohumil Fafilek (iD) http://orcid.org/0000-0002-3869-7937
Pavel Krejci (iD) http://orcid.org/0000-0003-0618-9134
Kalina Hristova (iD) http://orcid.org/0000-0003-4274-4406

Joint Public Review: https://doi.org/10.7554/eLife.88144.4.sa1

Author Response https://doi.org/10.7554/eLife.88144.4.sa2

## Additional files

### Supplementary files
- Supplementary file 1. Best-fit Gaussian parameters for the different log(brightness) distributions.
- Supplementary file 2. Calculated Z values from the Gaussian fit parameters in *Supplementary file 1*.
- Supplementary file 3. $\beta'_4$, $\beta'_8$, and $\beta'_9$ values calculated for each response using *Equation 4*.
- Supplementary file 4. Comparison of $\beta'_4$, $\beta'_8$, and $\beta'_9$ values from *Supplementary file 3*.
- Supplementary file 5. Summary of results shown in *Figure 5A–C*.
- MDAR checklist

### Data availability
All data are included in the manuscript. All source data have been provided.

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
