## [Editor Report · eLife assessment]

This manuscript describes **useful** data on the mechanisms underlying the activation of the receptor tyrosine kinase FGFR1 and stimulation of intracellular signaling pathways in response to FGF4, FGF8, or FGF9 binding to the extracellular domain of FGFR1. **Solid** evidence for quantitative differences in the downstream responses induced by the three ligands is presented. This manuscript will be of interest to biochemists and cell biologists working on receptor tyrosine kinases and general cell signalling across membranes.

---

## [Referee Report · Joint Public Review]

In this manuscript, Karl et al. explore mechanisms underlying the activation of the receptor tyrosine kinase FGFR1 and stimulation of intracellular signaling pathways in response to FGF4, FGF8, or FGF9 binding to the extracellular domain of FGFR1. The manuscript demonstrates that FGF4, FGF8, and FGF9 exhibit distinct binding modes towards FGFRs. It is also proposed that FGF8 exhibits "biased ligand" characteristics that is manifested via binding and activation FGFR1 mediated by unproven and speculative "structural differences in the FGF-FGFR1 dimers, which impact the interactions of the FGFR1 trans membrane helices, leading to differential recruitment and activation of the downstream signaling adapter FRS2".

---

## [Author Response]

The following is the authors’ response to the current reviews.

Comment. “The manuscript demonstrates that FGF4, FGF8, and FGF9 exhibit distinct binding modes towards FGFRs”

No, this paper is not about ligand binding, and there are NO binding data in the manuscript. This paper is about ligand-dependent functional bias. Previously, differential effects of ligands on the signaling of one FGFR have been attributed to differences in ligand binding, but that paradigm is incomplete, if not incorrect. This manuscript is the first demonstration that three FGF ligands induce bias in FGFR1 signaling. FGF8 preferentially activates some of the probed downstream responses (FRS2 phosphorylation and extracellular matrix loss), while FGF4 and FGF9 preferentially activate different probed responses (FGFR1 phosphorylation and growth arrest). The bias we report here cannot be the result of differences in ligand binding. Indeed, if the differences between ligands are only in the binding strength, then a strongly binding ligand at low concentration will act identically to weakly binding ligand at high concentration. Our article thus changes the current paradigm about how FGF ligands activate FGFR signaling.

Comment. It is also proposed that FGF8 exhibits "biased ligand" characteristics.

We do not “propose” the existence of ligand bias, we demonstrate it in the manuscript by following the latest IUPHAR community guidelines on bias identification and quantification (Kolb et al, 2022). We calculate bias coefficients, and we analyze the results using statistical tools.

Comment. …“Unproven and speculative structural differences in the FGF-FGFR1 dimers”.

This statement is not correct, as it is directly contradicted by the differences reported in Figure 6. This Figure presents the results of a quantitative FRET assay performed at high ligand concentration, which ensures that there are no monomeric receptors. Under these conditions, the measured FRET efficiency depends only on the dimer conformation. The measured differences in FRET efficiencies reveal distinct differences in the FGFR1 TM domain dimer conformations when FGF8 is bound to the extracellular domain of FGFR1, as compared to FGF4 and FGF9. The difference can be observed in the raw FRET data in Figure 6A. While these data do not reveal the exact molecular origin of the structural differences, they unequivocally prove that there are structural differences when different ligands are bound.

References

Kolb P, Kenakin T, Alexander SPH, Bermudez M, et al. Community guidelines for GPCR ligand bias: IUPHAR review 32. Br J Pharmacol. 2022;179, 3651-3674.

The following is the authors’ response to the previous reviews.

eLife assessment. This manuscript describes useful data on the mechanisms underlying the activation of the receptor tyrosine kinase FGFR1 and stimulation of intracellular signaling pathways in response to FGF4, FGF8, or FGF9 binding to the extracellular domain of FGFR1. Solid quantitative binding experiments are presented to demonstrate that FGF4, FGF8, and FGF9 exhibit distinct binding affinities towards FGFRs.

No, this paper is not about binding, and there is NO binding data in the manuscript. This paper is about function. This is the first demonstration that three FGF ligands induce bias in FGFR1 signaling. Thus far, differential effects in the signaling of one FGFR have been attributed to differences in ligand binding, but this current paradigm is incomplete/incorrect. Our article changes the current paradigm in how FGF activate downstream FGFR signaling.

We have clarified this point by adding the following text in the Discussion.

"Thus far, differential effects in the signaling of one FGFR in response to different FGF ligands have been attributed to differences in ligand binding. It can be reasoned, however, that differences in ligand binding strengths, alone, cannot explain differential signaling. Indeed, if the differences between ligands are only in the binding strength, then a strongly binding ligand at low concentration will act identically to weakly binding ligand at high concentration. Here we discovered, using tools that are novel for the RTK field, that there are qualitative differences in the actions of the ligands. FGF8 preferentially activates some of the probed downstream responses (FRS2 phosphorylation and collagen loss), while FGF4 and FGF9 preferentially activate different probed responses (FGFR1 phosphorylation and growth arrest). These effects occur in addition to previously measured differences in ligand binding coefficients (87).”

We have also re-written the abstract.

“Abstract

“The mechanism of differential signaling of multiple FGF ligands through a single FGF receptor is poorly understood. Here, we use biophysical tools to quantify multiple aspects of FGFR1 signaling in response to FGF4, FGF8 and FGF9: potency, efficacy, bias, ligand-induced oligomerization and downregulation, and conformation of the active FGFR1 dimers. We find that the three ligands exhibit distinctly different potencies and efficacies for inducing responses in cells. We further discover qualitative differences in the actions of the three FGFs through FGFR1, as FGF8 preferentially activates some of the probed downstream responses (FRS2 phosphorylation and extracellular matrix loss), while FGF4 and FGF9 preferentially activate different probed responses (FGFR1 phosphorylation and cell growth arrest). Thus, FGF8 is a biased ligand, when compared to FGF4 and FGF9. Förster resonance energy transfer experiments reveal a correlation between biased signaling and the conformation of the FGFR1 transmembrane domain dimer. Our findings expand the mechanistic understanding of FGF signaling during development and bring the poorly understood concept of receptor tyrosine kinase ligand bias into the spotlight.”

**Reviewer #1 (Public Review):**
Comment. Quantitative binding experiments presented in the manuscript demonstrate that FGF4, FGF8, and FGF9 exhibit distinct binding affinities towards FGFRs.

This paper is not about binding, and there is NO binding data in the manuscript. This paper is about function. Please see our response to the Elife assessment.

Comment. It is also proposed that FGF8 exhibits "biased ligand" characteristics that is manifested via binding and activation FGFR1 mediated by "structural differences in the FGF- FGFR1 dimers, which impact the interactions of the FGFR1 transmembrane helices, leading to differential recruitment and activation of the downstream signaling adapter FRS2".

We do not “propose” the existence of ligand bias, we demonstrate it in the manuscript by following the latest IUPHAR community guidelines on bias identification and quantification (Kolb et al, 2022). Specifically, we construct bias plots, we calculate bias coefficients, and we analyze the results using statistical tools.

Also, please note that ligand bias has no direct connection to binding strength, so the statement that biased ligand characteristics “is manifested via binding” is not correct.

Comment. In the absence of any structural experimental data of different forms of FGFR dimers stimulated by FGF ligands the model presents in the manuscript is speculative and misleading.

Figure 6 presents the “structural experimental data”. A quantitative FRET assay is performed at high ligand concentration, which ensures that there are no monomeric receptors. Under these conditions, the measured FRET efficiency depends only on the dimer conformation. The measured FRET efficiencies reveal distinct differences in the FGFR1 TM domain dimer conformations when the ligand FGF8 is bound to the extracellular domain of FGFR1, as compared to the cases of FGF4 and FGF8.

Because the Rosetta modeling of the kinase domains in the previous version of the paper is not based on experimental data, we have removed the modeling from the Results, and we have removed all references to it in the Discussion. Thus, all that is shown and discussed in the revised paper is based on experimental data.

We have substituted two paragraphs in the discussion with the following two sentences:

“The experimental data in Figure 6 hint at the possibility that ligand bias arises due to differences in FGFR1 dimer conformations. If this is so, then conformational differences in the signaling complex in the plasma membrane underlie biased signaling for both RTKs and GPCRs, the two largest receptor families in the human genome”.

References

Kolb P, Kenakin T, Alexander SPH, Bermudez M, et al. Community guidelines for GPCR ligand bias: IUPHAR review 32. Br J Pharmacol. 2022;179, 3651-3674.